



# Imaging structure and geometry of slabs in the greater Alpine area - A P-wave traveltime tomography using AlpArray Seismic Network data

Paffrath Marcel[1], Friederich Wolfgang[1], and the AlpArray and AlpArray-Swath D Working Group [*]

[1]Ruhr-Universität Bochum
[1]For further information regarding the team, please visit the link which appears at the end of the paper.

**Correspondence:** Paffrath M. (marcel.paffrath@rub.de)

**Abstract.** We perform a teleseismic P-wave traveltime tomography to examine the geometry and structure of subducted lithosphere in the upper mantle beneath the Alpine orogen. The tomography is based on waveforms recorded at over 600 temporary and permanent broadband stations of the dense AlpArray Seismic Network deployed by 24 different European institutions in the greater Alpine region, reaching from the Massif Central to the Pannonian Basin and from the Po plain to the river Main.

Teleseismic traveltimes and traveltime residuals of direct teleseismic P-waves from 331 teleseismic events of magnitude 5.5 and higher recorded between 2015 and 2019 by the AlpArray Seismic Network are extracted from the recorded waveforms using a combination of automatic picking, beamforming and cross-correlation. The resulting database contains over 162.000 highly accurate absolute P-wave traveltimes and traveltime residuals. For tomographic inversion, we define a model domain encompassing the entire Alpine region down to a depth of 600 km. Outside this domain, a laterally homogeneous standard

earth model is assumed. Predictions of traveltimes are computed in a hybrid way applying a fast Tau-P method outside the model domain and continuing the wavefronts into the model domain using a fast marching method. For teleseismic inversion, we iteratively invert demeaned traveltime residuals for P-wave velocities in the model domain using a regular discretization with an average lateral spacing of about $25\,\mathrm{km}$ and a vertical spacing of $15\,\mathrm{km}$. The inversion is regularized towards an initial model constructed from an a priori model of the crust and uppermost mantle and a standard earth model beneath.

The resulting model provides a detailed image of slab configuration beneath the Alpine and Apenninic orogens. Major features are an overturned Adriatic slab beneath the Apennines reaching down to $400\,\mathrm{km}$ depth still attached in its northern part to the crust but exhibiting detachment towards the southeast. A fast anomaly beneath the western Alps indicates a short western Alpine slab that ends at about $100\,\mathrm{km}$ depth close to the Penninic front. Further to the east and following the arcuate shape of the western Periadriatic Fault System, a deep-reaching coherent fast anomaly with complex interior stucture generally

dipping to the SE down to about $400\,\mathrm{km}$ suggests a slab of European origin extending eastward to the Giudicarie fault. This slab is detached from overlying lithosphere at its eastern end below a depth of about $100\,\mathrm{km}$. Further to the east, well-separated from the slab beneath the western and central Alps, another deep-reaching, nearly vertically dipping high-velocity anomaly suggests the existence of a slab beneath the Eastern Alps of presumably European origin which is completely detached from the orogenic root. Our image of this slab does not require a polarity switch because of its nearly vertical dip and full detachment

from the overlying lithosphere. Fast anomalies beneath the Dinarides are weak and concentrated to the northernmost part and





shallow depths. Low-velocity regions surrounding the fast anomalies beneath the Alps to the west and northwest follow the same dipping trend as the overlying fast ones, indicating a kinematically coherent subducting tectosphere in this region. In contrast, low-velocity anomalies to the east suggest asthenospheric upwelling presumably driven by retreat of the Carpathian slab and extrusion of eastern Alpine lithosphere towards the east while low velocities to the south are presumably evidence of
asthenospheric upwelling and mantle hydration due to the backarc position behind the European slab.

## 1   Introduction

For well over a century, the Alps have been the breeding ground for revolutionary ideas in geodynamics that serve as blueprints for understanding other major orogens in the world. Path-breaking concepts such as nappes tectonics and cylindrism, and fundamental discoveries such as ophiolites and ultra-high-pressure metamorphic rocks have their origin in the Alps. Being
in a transition from subduction to continental collision, the Alps exhibit the entire spectrum of orogenic processes, from juvenile stages in their eastern part to mature stages in the central and western regions. The Alps are marked by highly dynamic small-scale interactions of microplates in a setting of Europe-Africa convergence which produced their unique arcuate, non-cylindrical shape and lead to characteristic along-strike changes of indentation. In spite of their small size, they exhibit significant topographic relief. Several regions in the Alps (e.g. Friaul) are prone to considerable seismic hazard and, owing to
the high population density, suffer from high vulnerability to natural disasters.

    Due to these unique characteristics, the Alps have been the subject of continuous seismological research. Crustal structure was intensively studied using active seismic refraction and reflection surveys (Blundell et al., 1992; Mueller and Banda, 1983; Hirn et al., 1989; Pfiffner et al., 1988; Frei et al., 1989; TRANSALP Working Group et al., 2002). On top of that, crustal velocities and Moho topography were investigated using local earthquake tomography (Diehl et al., 2009), receiver func-
tions (Lombardi et al., 2008; Kummerow et al., 2004) and ambient noise tomography (Fry et al., 2010; Molinari et al., 2015). Models of mantle P-wave velocity structure were derived from teleseismic body wave tomography both on regional scale (Lippitsch et al., 2003; Mitterbauer et al., 2011; Zhao et al., 2016) and continental scale (e.g., Spakman et al., 1993; Piromallo and Morelli, 2003). First concepts of mantle flow beneath the Western Alps were developed from results of shear-wave splitting analysis (Barruol et al., 2011) while topography of the mantle discontinuities was mapped using receiver functions (Lom-
bardi et al., 2009). Continental-scale models of mantle S-wave velocities were inferred from surface waves either by inversion of waveforms (Marone et al., 2004; Legendre et al., 2012) or dispersion curves (Peter et al., 2008; Boschi et al., 2009; El-Sharkawy et al., 2020), from S-wave traveltimes (Schmid et al., 2008) and from adjoint full waveform inversion (Zhu et al., 2012). Recently, Kästle et al. (2018) derived a regional-scale S-wave model of the greater Alpine region by combining surface wave dispersion extracted from earthquake and ambient noise data.
The current picture of the deep structure of the Alps that can be drawn from the seismological results looks as follows: The crust is characterized by stacking and wedging of tectonic units of European and Adriatic provenience to accomodate crustal shortening during collision. The Moho is discontinuous with gaps as well as vertical offsets between European and Adriatic Moho. The mantle below is pervaded by distorted, corrugated and torn slab segments of presumably continental origin whose





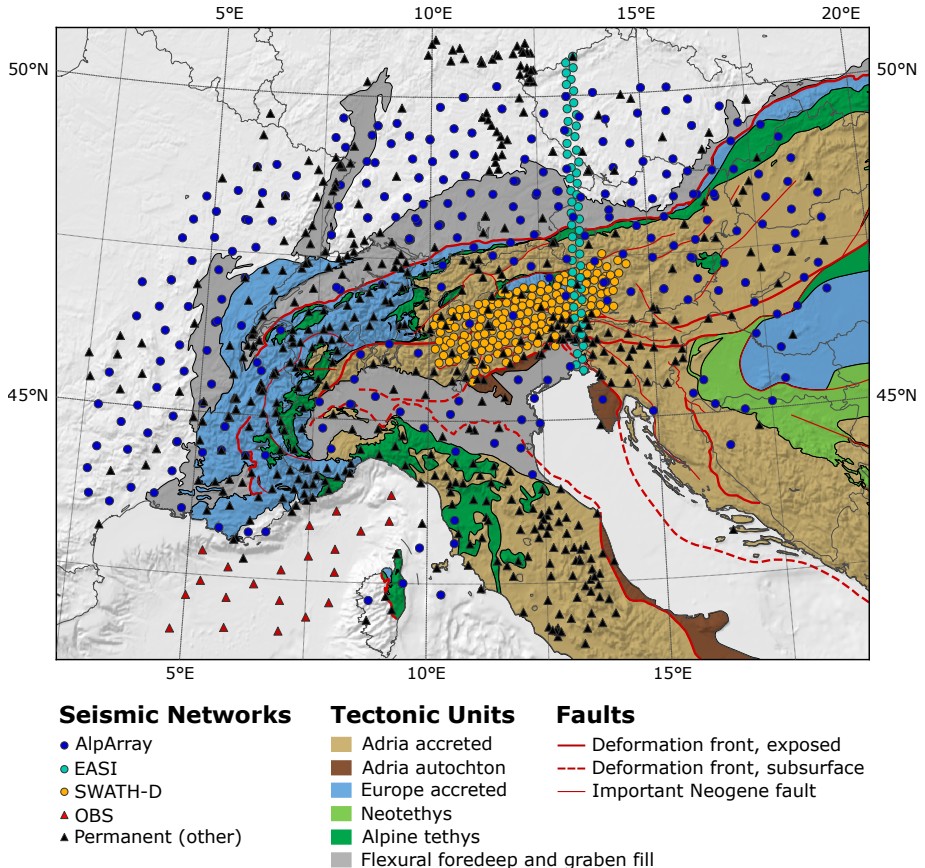

**Figure 1.** Tectonic map of the Alpine chains compiled by M.R. Handy showing different seismic networks and temporary experiments, tectonic units and major fault systems.

dip apparently flips from SE in the western to central parts to NE in the eastern part of the Alps. This postulated polarity flip is

a unique feature of Alpine orogeny and has not been observed elsewhere in the world. Further east, in the northern Dinarides, a slab seems to be entirely missing. Deep in the transition zone, the entire Alpine belt and the southern forelands are underlain by seismically fast material interpreted as remnants of the Alpine Tethys subducted during Late Cretaceous to Early Paleogene.

Despite the successes of Alpine geophysical research, the current state of knowledge on the deep structure of the Alps cannot be considered as satisfactory given major open questions and new opportunities of seismological observation and imaging. A

clear definition of the structural properties of the lithosphere is still lacking. How do slabs behave in the shallower regions? What is the nature of coupling of the slabs with the plates at the surface? How does it influence surface topography? Are parts of the subducted slabs still attached to the plate. Which parts (lower crust, lithospheric mantle) have sunk into the mantle? Images of oppositely dipping slab segments as well as their interpretation are still controversial. Is European or Adriatic lithosphere subducted in the Eastern Alps?





The internal structure, outer boundaries and provenance of subducted material are poorly constrained. Suggested slab tears in the Western and Central Alps (Kästle et al., 2020) remain enigmatic. Up to now, resolution is concentrated on the core region of the Alps, hence extensions as well as connections of subducted material to surrounding lithosphere and asthenosphere are unclear. Which role did the old Variscan basement play during Alpine orogeny? To what extent is lithospheric structure inherited from this older history? Finally, how thick and how long are the descending slabs? Where are the younger slabs

connected to the older subducted material stacked in the transition zone? How much subducted material has accumulated in the transition zone? An answer to these questions is essential for plate tectonic reconstructions and geodynamic modelling.

In this paper, we attempt to provide answers to most of the questions formulated above by means of a teleseismic tomography of the mantle beneath the entire Alpine mountain belt and its forelands. The observational basis for this study are the AlpArray Seismic Network (AASN, Hetényi et al., 2018), the complementary experiments EASI (Eastern Alpine Seismic Investigation,

EASI, 2014) and Swath-D (Heit et al., 2017) dedicated to specific target regions in the Alps, and a joint German-French ocean bottom seismometer deployment in the Ligurian sea (LOBSTER and ALP-ARRAY-FR). The AASN covers the entire Alpine mountain belt including the forelands from the Massif Central in the west to the Pannonian Basin in the east, and from the river Main in Germany in the north to the Po river in the south. In its full configuration, it encompassed about 600 seismic broadband stations (Fig 1). Compared to the stations available for previous body wave tomographic images (Lippitsch et al.,

2003; Mitterbauer et al., 2011; Zhao et al., 2016) which form the basis for our current view of the mantle beneath the Alps, the AASN constitutes a massive improvement of observational coverage. The AASN is the densest seismic network that ever covered an entire mountain belt and permits unprecedented images of the 3D distribution of seismic velocities in the lithosphere and the sub-lithospheric mantle.

In section 2, we sketch the methods applied to extract traveltimes and traveltime residuals from the original waveform data

and to make predictions of traveltimes for teleseismic P-wave phases. Moreover, we describe the inversion procedure and our approach of implementing the effect of crustal structure into the inversion procedure. Section 3 describes the data basis of this study in a brief manner as a detailed analysis of the traveltime data is already provided by Paffrath et al. (2021). Section 4 gives an analysis of resolution by means of checkerboard tests and in section 5 we present the tomographic velocity model based on horizontal and selected vertical cross sections. The discussion in section 6 gives an interpretation of the structural significance

of the observed velocity anomalies, in particular focusing on the geometry of subducted lithosphere beneath the Alps. Our results are compared to previous tomographies by Lippitsch et al. (2003), Mitterbauer et al. (2011) and Zhao et al. (2016). We conclude with a 3-dimensional image of the slab configuration beneath the Alps directly derived from the tomographic model.

## 2  Methods

### 2.1  Measuring traveltimes and traveltime residuals

We define here traveltime $\tau_j$ as the difference between the measured arrival time of the P-wave at some station $j$ and the origin time of the considered earthquake. For the latter, we use the centroid time given in the global CMT catalogue (Dziewonski et al., 1981; Ekström et al., 2012).





The process of determining traveltimes is described in detail by Paffrath et al. (2021). A specially designed combination of automatic picking, waveform cross-correlation and beamforming is applied to attain the accuracy necessary for performing tele-
seismic tomography. Initial automatic picks based on higher-order statistics and the Akaike information criterion (Küperkoch et al., 2010) are used as anchor points for a first stage cross-correlation with an appropriately selected reference station. Traces are then aligned to the reference trace and stacked to form a very low noise beam trace. The beam trace serves as reference trace for a second cross-correlation step in which time differences $\Delta\tau_{jB}$ between the beam and the individual stations are measured. Absolute traveltime $\tau_j$ is determined by automatically picking the beam trace, $\tau_B$, and adding the time difference with respect
to the beam:

$$\tau_j = \tau_B + \Delta\tau_{jB}. \tag{1}$$

In the teleseismic inversion described below, we do not use these absolute traveltimes but instead invert event-wise demeaned traveltime residuals for velocity perturbations. Demeaning is done to avoid errors of origin times and to reduce influences from remote earth structure not included in the inversion domain. Residuals are obtained by subtracting theoretical traveltimes of an
appropriate 1D reference earth model from the observed traveltimes. Thus, the demeaned traveltime residual can be written as:

$$r_j = \tau_j - \overline{\tau} - (T_j - \overline{T}), \tag{2}$$

where $\overline{\tau}$ denotes the observed traveltime array average, $T_j$ the reference earth traveltimes and $\overline{T}$ their array average. Inserting eq. (1), we find

$$r_j = \tau_B + \Delta\tau_{jB} - \tau_B - \overline{\Delta\tau} - (T_j - \overline{T})$$
$$= \Delta\tau_{jB} - \overline{\Delta\tau} - (T_j - \overline{T}), \tag{3}$$

where $\overline{\Delta\tau}$ denotes the array average of the time difference between station traces and beam.

Remarkably, the demeaned residual traveltimes can be calculated from the measured time differences between station trace and beam alone, and hence, their uncertainty is controlled by the uncertainty of these time differences. We estimate this uncertainty on the basis of the full width of the cross-correlation function at half maximum, FWHM, and its normalized
maximum value $C_{\mathrm{max,jB}}$:

$$\sigma_j = \mathrm{FWHM}_{jB}\left(1 - C_{\mathrm{max,jB}}\right). \tag{4}$$

This results in an uncertainty which tends to become larger with decreasing waveform frequency since cross-correlation maxima become wider. Moreover, waveforms deviating from the beam trace due to noise or additional complexity receive a larger uncertainty owing to a reduced maximum correlation.

## 2.2 Computation of synthetic traveltimes

Teleseismic tomography requires the computation of synthetic traveltimes for comparison with the observed ones. We compute traveltimes in a hybrid way using a fast method for the path from the source location to the inversion domain assumed to lie





within a 1D standard earth model and a 3D eikonal solver for tracing the wavefront through the inversion domain. Since the seismic velocity only changes in the inversion domain, rays and traveltimes through the standard earth model are only computed once for each event whereas traveltimes in the inversion domain are recalculated in each iteration of the inversion. We use the TauP routine (Crotwell et al., 1999) implemented in ObsPy (Krischer et al., 2015) to calculate ray propagation in the global earth. The ray parameters and traveltimes are saved on each boundary surface exposed to the source. From there, the incident wavefront is continued into the three-dimensional model using the FM3D algorithm by Rawlinson and Sambridge (2005).

The influence of global heterogeneities on rays between a given source and the array is expected to be low, as rays are bundled for the largest part of the travel path and only spread into different directions when approaching the inversion domain.Thus, the influence of remote heterogeneities will mostly affect all rays from a given source in the same way and is eliminated when subtracting the array average. Moreover, we assume the earth's lower mantle to be weakly heterogeneous in P-wave velocity (e.g., Aki et al., 1977) in contrast to the upper mantle and crust below the Alps. However, there will be remaining influences of the earth's lower mantle on our model.

## 2.3 Inversion Method

The inverse problem is solved using the inversion scheme implemented in FMTOMO. Demeaned traveltime residuals as defined in eq. (2) are inverted for P-wave velocity, minimizing the objective functional (e.g. Rawlinson et al., 2006):

$$
\begin{aligned}
S(m) =& (g(m) - d_{obs})^T C_d^{-1} (g(m) - d_{obs}) \\
& + \epsilon (m - m_0)^T C_m^{-1} (m - m_0) \\
& + \eta (m - m_0)^T D^T D (m - m_0)
\end{aligned}
\tag{5}
$$

with $m$ denoting the 3D P-wave velocity model, $m_0$ the initial P-wave velocity model, $g(m)$ predicted, $d_{obs}$ observed demeaned residuals and $C_d$ an a priori data covariance matrix representing the uncertainties of the residuals. The first term represents the misfit between observations and predictions while the two following terms define the squared model norm which is a measure of the complexity of the model. The first term of the model norm describes the variance of the model with respect to the initial model scaled by a piori variances contained in the matrix $C_m^{-1}$ while the second term quantifies the roughness of the model using the matrix $D$ which implements a 3D Laplacian differential operator. The relative weight of variance, roughness and misfit in the objective function can be controlled by the parameters $\epsilon$ and $\eta$. Misfit and model norm play against each other: reducing the misfit implies increasing the model norm and vice versa. The aim of the inversion is to find a model that reduces the misfit to a certain threshold and mimizes the model norm.

Variance and roughness parameters are selected with the help of synthetic tests, where we investigate the recovery of a known input model after a fixed number of iterations and with the help of trade-off curves by which we try to balance out misfit and model norm. We will show such a curve in Fig. 11 when analysing the actual trade-off for different variance and roughness parameters in Sect. 5. The ideal trade-off is found on the curve somewhere close to the maximum curvature.

The inverse problem is non-linear because the forward problem $g(m)$ is a non-linear functional. For this reason, the minimum of the objective functional is searched iteratively. In each iteration, the linearised inverse problem is solved using a subspace



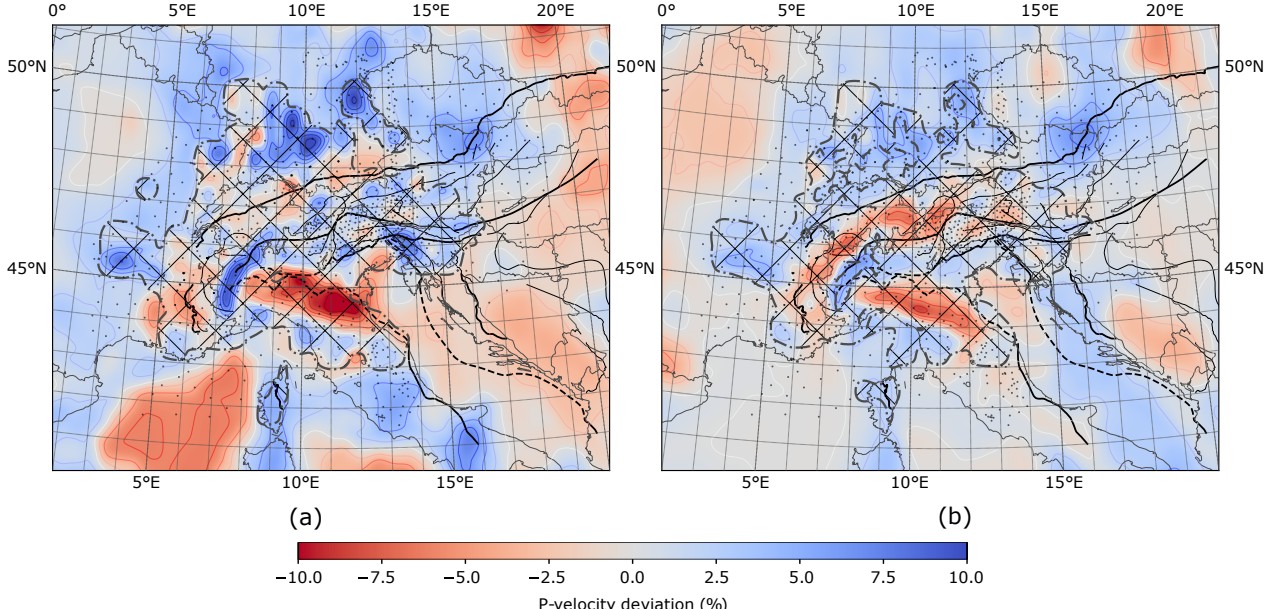

(a)         (b)

P-velocity deviation (%)

**Figure 2.** Two horizontal slices through the (smoothed) crustal model at (a) $15\,\mathrm{km}$ and (b) $30\,\mathrm{km}$ depth. Colors show the difference in P-wave velocity to the one-dimensional background model. Dash-dotted line indicates boundary between the two models of Diehl et al. (cross-hatched area) and Tesauro et al..

projection method and the predicted traveltimes $g(m)$ are updated using the fast-marching method (Rawlinson and Sambridge, 2005). Iterations are stopped if the misfit over number of data defined by

$$\chi^2/N = \frac{1}{N}(g(m) - d_{obs})^T C_d^{-1} (g(m) - d_{obs}) \tag{6}$$

either reaches 1, signifying that the data are fit down to their uncertainites, or if misfit reduction stagnates when executing
further iterations.

    The model is discretized by defining the model parameters $m$ as the values of P-wave velocity on a regular grid with a spacing of $0.23°$ in latitude, $0.3°$ in longitude and $15\,\mathrm{km}$ in depth. This corresponds to a lateral grid spacing of about $25\,\mathrm{km}$. The bounds of the model domain are at $31°$ and $61°$ in latitude, $-13°$ and $35°$ in longitude and at a depth of 600 km. The center of the inversion domain is at $46°$ latitude and $11°$ longitude. The total number of grid points and thus model parameters
is $884.520$. The lateral dimensions of the computational box were chosen larger than actually necessary for the inversion. Due to numerical errors of up to $0.3\,\mathrm{s}$ in the synthetic traveltimes near the lateral boundaries of the model domain when calculated with the hybrid method, we have increased the model domain to ensure that traveltime errors in the relevant areas of the model stay below $0.01$ s. For consistency, we also used the hybrid method for the calculation of the teleseismic reference traveltimes.





## 2.4 Integration of a priori high resolution models of the crust and uppermost mantle

The observed traveltime residuals contain contributions from mantle and crustal heterogeneities. Unfortunately, resolution of crustal structures by teleseismic waves is very limited. Owing to the subvertical propagation of teleseismic waves through the crust, crossing of rays is essentially inhibited leading to a complete loss of vertical resolution. Hence, to infer mantle structure, an a priori velocity model of the crust based on independent data is required (e.g., Kissling, 1993).

While the standard approach is either a direct correction of the observed traveltime residuals at each station assuming vertical
wave propagation through the crust (e.g., Zhao et al., 2016), or an event-wise calculation of crustal residuals (e.g., Lippitsch et al., 2003), we directly integrate the external crustal model into the velocity model of our teleseismic inversion. In this way, ray refraction in the crust depending on vertical and azimuthal incidence is correctly treated. Moreover, we may use the variance term of the model norm to assign spatially varying variances to velocities in the crustal region that reflect the reliability of the external crustal model. This allows the inversion to deviate from the a priori crustal velocities if required by the teleseismic
data. Finally, with this method, we obtain a velocity model covering crust and mantle that consistently reconciles imported crustal heterogeneities and teleseismic data.

We have assembled a new P-wave velocity model of the crust and uppermost mantle beneath the greater Alpine region (Fig. 2) mainly based on the model EuCRUST-07 by Tesauro et al. (2008) and the fully three-dimensional, high resolution P-wave velocity model of the central Alpine crust provided to us by T. Diehl (Diehl et al., 2009). EuCRUST-07 is discretized
laterally on a $15'$ times $15'$ regular grid. Vertically, each grid cell contains three layers composed of sediments, upper and lower crust. Values of thickness and seismic velocities are specified for the crustal layers while only thickness is given for the sediment layer. Here, we use a constant value of $2.8$ km/s for the P-wave velocity of the sediments. EuCRUST-07 was compiled from results of various seismic reflection, refraction and receiver function studies. The model by Diehl was derived using local earthquake tomography and provides the most precise information on crustal (and uppermost mantle) velocity
structure. However, it only covers about two third of the lateral dimensions of AlpArray and has varying resolution that reflects the available distribution of local earthquakes and seismic stations. Furthermore, we take advantage of a more recent study of Spada et al. (2013), which focuses on the greater Alpine area, to update Moho depth and hence crustal thickness of the EuCRUST-07 model.

To construct the a priori crustal model, we discretized the modified EuCRUST-07 model on a regular $2 \times 2 \times 2$ km grid.
In the regions where Diehl's model provides P-wave velocity values, we form a linear combination of both models with weights reflecting the reliability of Diehl's model. The weights are calculated on the basis of the diagonal elements of the resolution matrix (RDE) provided to us by T. Diehl (personal communication). The weight is calculated with a transfer function (Fig. 3), favoring the sufficiently well resolved parts of the model of Diehl et al. over that of Tesauro et al.. Finally, since the discretization of the inversion domain is much coarser than the $2 \times 2 \times 2$ km grid, we resample the velocitiy values of our a
priori crustal and uppermost mantle model using a 3-D Gaussian kernel as a spatial filter to prevent aliasing effects.

The a priori crustal model serves as initial model for the inversion as described in the previous section. For the regions of the inversion domain beyond the extent of the a priori crustal model, we assume a modified standard earth model (dotted line





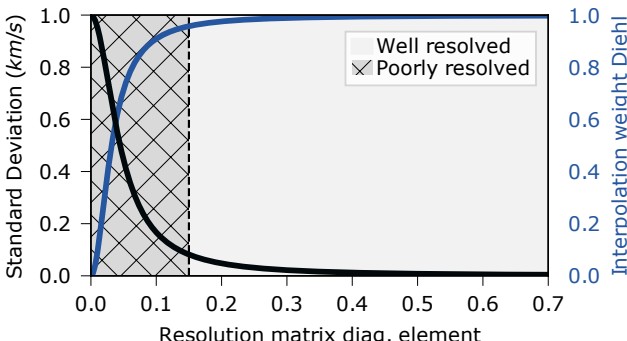

**Figure 3.** Black line: Transfer function mapping RDE values provided by Diehl to a priori standard deviations of P-wave velocity. Blue line: Transfer function mapping RDE values to interpolation weights for Diehl's model.

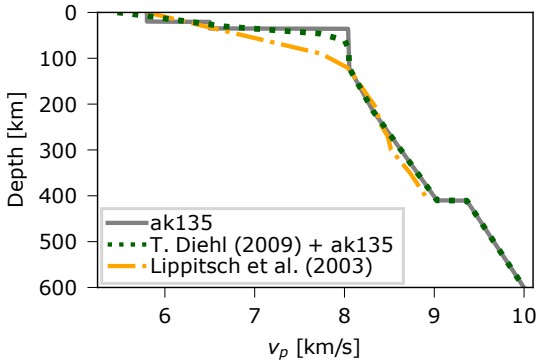

**Figure 4.** Dotted line: One-dimensional model used in this study as initial model for the inversion in regions beyond the extent of the a priori crustal model. For comparison, the standard Earth model AK135 (solid line) and the reference model used by Lippitsch et al. (2003) (dash-dotted line).

in Fig. 4) constructed from the standard earth model AK135 (Kennett et al., 1995) and the 1D reference model used by Diehl et al. (2009). The transitions from the a priori crustal model to this model occurs at a depth of $77.5\,\mathrm{km}$.

We also estimate a priori standard deviations for different parts of the crustal model which control the amount of adjustments of P-wave velocity allowed during teleseismic inversion. For regions in our crustal model controlled by the model of Diehl et al., we use again an empirical transfer function to calculate a standard deviation for a given RDE value (Fig. 3). In the remaining regions, where we use the model of Tesauro et al., we assume an equivalent RDE of $0.1$ for the sedimentary basins and the lower crust and $0.15$ for the upper crust corresponding to a standard deviation of $0.17\,\mathrm{km/s}$ and $0.082\,\mathrm{km/s}$, respectively. It should

be noted here that the effect of our a priori crustal model on the inversion result is additionally influenced by the damping parameter $\epsilon$. For this reason, it is rather the ratios of the standard deviations than their absolute values that matter.



## 3    Data Basis

As already mentioned, this study is founded on data from the AlpArray Seismic Network (AASN, Hetényi et al., 2018), the complementary experiments EASI (Eastern Alpine Seismic Investigation, EASI, 2014) and Swath-D (Heit et al., 2017), and
a joint German-French ocean bottom seismometer deployment in the Ligurian sea (LOBSTER and ALP-ARRAY-FR). In its full extent, the network encompassed about 600 seismic broadband stations of which about one half are permanently operating stations and the other half temporarily deployed stations (Fig 1).

Installation of stations began in July 2015 mainly in Austria and Northern Italy, followed by stations covering southern Germany in November/December 2015 and eastern France in summer 2016. Ocean bottom seismometers were deployed from
June 2017 to February 2018. The earliest complementary experiment, partly included in our dataset, is the EASI project with 55 stations deployed on a north south profile at 13.4°E traversing the Alps from the northern Alpine foreland to the Adriatic Sea and recording ground motion for more than a year until August 2015. The second complementary experiment, Swath-D, running for 2 years from the end of 2017 and covering a broad swath across the eastern Alps including the Tauern window, further improved station density in a key area of the Alps.

To further extend the station coverage to the north and south, we include all permanent and temporary stations available at the time of request in the greater Alpine region, within a 5° radius around 46° N and 11° E, i.e. mainly stations in central Germany and northern Italy, into our dataset. With that we finally assemble waveforms from a total of 1025 different seismic broadband stations with recording times scattered through a period of over four and a half years between 2015 and the end of 2019, with a peak in station coverage of more than 720 stations in late 2017.

We give here only a brief overview of the traveltime database that was extracted from the recorded waveforms. For more detailed information, we refer the reader to Paffrath et al. (2021) who provide an in-depth analysis of the measured traveltimes and traveltime residuals and their errors. In that study, traveltimes were measured on waveforms filtered to two different maximum frequencies of 0.1 Hz and 0.5 Hz, respectively. For the present study, we invert traveltime residuals obtained from the higher frequency waveforms filtered to 0.5 Hz because they exhibit smaller uncertainties and thus higher spatial resolution.
The resulting database comprises 162366 traveltimes of 331 events of magnitude 5.5 or higher.

Azimuthal distribution of events is strongly inhomogeneous (Fig. 5), with the majority of events coming from northeastern and western directions. There is a lack of events especially from southern directions, as well as from southeast and northwest. This bias in azimuthal direction has to be taken into account when analysing results, as it might lead to smearing of different features in the direction of the least events, i.e. from nortwest to southeast, an effect we will evaluate in detail later in a
checkerboard test (Sect. 4).

Distances of events range from 35° to 135° to the centre of the array. Their distribution is also heterogeneous, scattering within only ±10° around 90° distance for backazimuths between 270° and 40° and for the full range of up to ±50° for the remaining backazimuths. However, we expect the influence of heterogeneity in distance distribution to be minor, as the maximum resulting difference in wavefront incident angles does not exceed 13° at the surface and 29° at 600 km depth for a





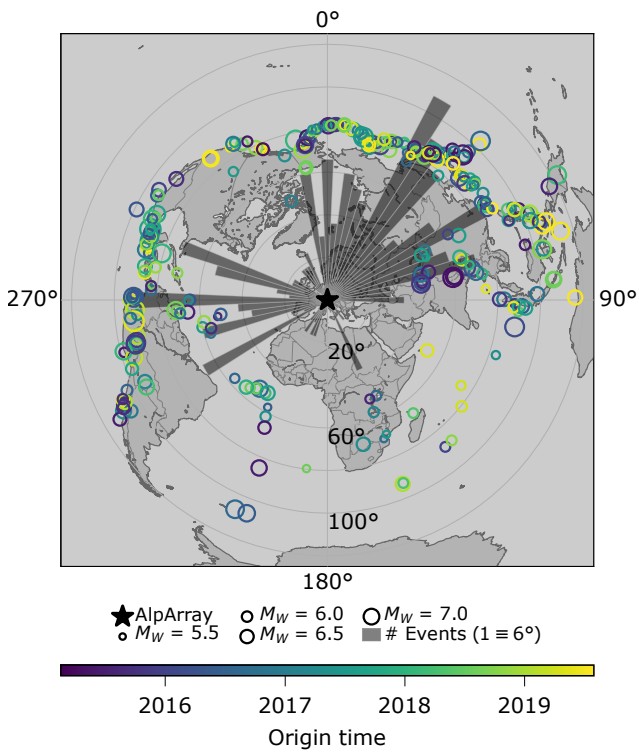

**Figure 5.** Distribution of 331 events of the high frequency dataset $db_{0.5}$. Size of circles correlates with moment magnitude, color with origin time. Histogram shows number of events binned in $5°$ bins azimuthally. A bar height of $60°$ radial distance equals to 10 events coming from that direction. The distribution is very irregular with most events located in the northeastern quadrant and in a western sector. There are large gaps with few or no events especially from the southeast as well as from the northwest. Peak value is 18 events for the back-azimuth interval between $30°$ and $35°$. The influence of the irregular distribution will get visible analysing resolution capabilities and smearing of the setup and can be seen especially well in the checkerboard test (Sect. 4)

1D earth model. Magnitude and origin time of the events do not show an azimuth or distance dependency except for a deficit of larger magnitude events in the southeast.

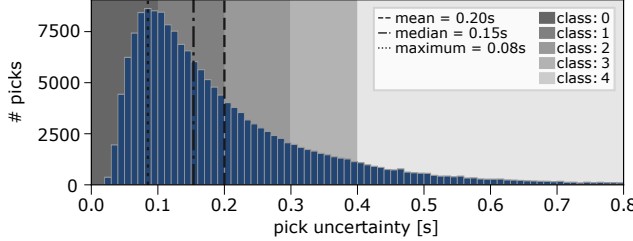

**Figure 6.** Histogram of uncertainties of the full dataset as shown in detail in (Paffrath et al., 2021). Mean and median of uncertainties are indicated by vertical lines, as well as the bin with the highest number of pick uncertainties.





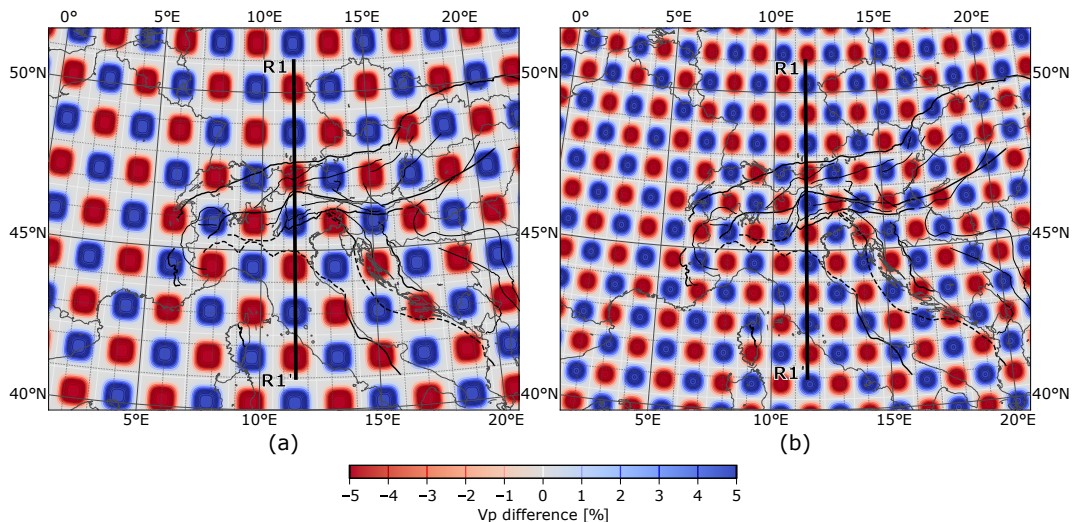

**Figure 7.** Horizontal slice cutting through the centre of the uppermost anomaly layer of the two checkerboard models. Depth of the first slice (a) of the $3 \times 3 \times 4$ model is $150\,\text{km}$. Depth of the slice through the $2 \times 2 \times 3$ model (b) is $175\,\text{km}$.

Traveltime uncertainties are categorized into five different classes in steps of $0.1\,\text{s}$ ranging from class 0 (best) below $0.1\,\text{s}$ to class 4 (worst), over $0.4\,\text{s}$. Although there is only a lower bound of the uncertainty for class 4, each onset in this class still has a well defined uncertainty and will be used for the tomography. The traveltime uncertainty distribution of the dataset

(Fig. 6) has a maximum at $0.08\,\text{s}$. The median of the distribution is $0.15\,\text{s}$, the mean is $0.2\,\text{s}$. Our average value of the estimated uncertainties is therefore lower than the one estimated by Lippitsch et al. (2003), who report a value of $0.36\,\text{s}$ for 4199 P-wave traveltimes and larger than the one reported by Zhao et al. (2016), who estimate a value of less than $0.1\,\text{s}$ for their 41838 traveltime residuals. However, due to the high number of stations in the AlpArray project, our dataset contains over 46.000 ($27\,\%$) of the onsets in the highest class with an estimated uncertainty $< 0.1\,\text{s}$. Less than $10\,\%$ are in the lowest class of $0.4\,\text{s}$ or

higher. A detailed analysis of pick uncertainties also regarding the regional distribution can be found in Paffrath et al. (2021).

## 4   Resolution Analysis

The computation of a formal resolution analysis was not possible due to the size of the inverse problem. Instead, we perform resolution tests based on two sets of test models exhibiting anomalies of alternating sign arranged in 3D checkerboard patterns of different size separated by cubes with zero anomaly to analyse smearing effects. The checkerboard domain starts below

the lower end of the a priori crustal and uppermost mantle model, i.e. at $100\,\text{km}$ depth. These perturbations are superimposed on the a priori crustal model and the 1D initial model. For all resolution tests, we fully account for our a priori crustal and uppermost mantle model when calculating synthetic traveltimes. Moreover, we add Gaussian noise to the synthetic traveltimes with a standard deviation given by the estimated uncertainties of the observed traveltimes. The inversion of the synthetic test data is performed in exactly the same way as the inversion of the observed data. The first checkerboard test model is made out





of tiles covering $3 \times 3 \times 4$ grid points resulting in an average anomaly size of ca. $75 \times 75 \times 60\,\mathrm{km}$. The second checkerboard pattern is made out of tiles covering $2 \times 2 \times 3$ grid points creating anomalies of ca. $50 \times 50 \times 45\,\mathrm{km}$ in size (Fig. 7b). Velocity perturbation is applied directly onto the inversion grid nodes, acting as a boxcar function. However, the spline interpolation in FMTOMO smoothes the edges of the anomalies, reducing the area of completely undisturbed space in between. For both patterns, the velocity perturbation reaches $\pm 10\,\%$ at the tile centre (Fig. 7).

Figure 8 shows horizontal slices of the recovered $3 \times 3 \times 4$ checkerboard pattern at different depths. The slices in the left column always cut through the centre of the anomalies and the slices in the right column cut through the unperturbed spaces in between the tiles. For the shallowest depth of $150\,\mathrm{km}$ (Fig. 8a), test model recovery within well resolved areas is excellent. However, there is a slight decrease of amplitude with distance from the centre of the array. Some smearing effects are visible that connect areas with the same sign of velocity perturbation, preferably along a NW-SE direction. We explain this behaviour

by the uneven event distribution (Sect. 3) that exhibits a strong deficit of earthquakes from SE and NW backazimuths leading to much less ray crossings along the NW-SE direction compared to the NE-SW direction. For this reason lateral resolution along the NW-SE direction is reduced compared to the NE-SW direction.

At a depth of $210\,\mathrm{km}$ (Fig. 8b), the test model perturbation vanishes. Nevertheless, a weak velocity perturbation appears after inversion indicating slight vertical smearing. Amplitudes in the gap are by almost one order of magnitude smaller than

at $150\,\mathrm{km}$ depth. At $270\,\mathrm{km}$ depth (Fig. 8c) the original checkerboard pattern is again clearly recovered, though we note a further decrease of the recovered amplitude. At $330\,\mathrm{km}$ depth (Fig. 8d) where the test model perturbation again vanishes, the recovered velocity perturbation is slightly stronger than at $210\,\mathrm{km}$ indicating a slight increase of smearing with depth. At $390\,\mathrm{km}$ depth (Fig. 8e), checkerboard recovery is still satisfactory but the recovered amplitude decreases further and NW-SE smearing becomes more prominent. The inverted velocity perturbation in the following zero perturbation layer of the test

model at $450\,\mathrm{km}$ depth (Fig. 8f) assumes amplitudes comparable to the full anomaly layer above pointing to a further increased smearing. Remarkably, at the greatest depth of $510\,\mathrm{km}$ (Fig. 8g) the recovery of the anomaly pattern is still excellent, even though the size of well recovered anomalies grows well beyond $75\,\mathrm{km}$ owing to lateral smearing.

The reconstruction results for the $2 \times 2 \times 3$ checkerboard resolution test (Fig. 9) are less favorable. In contrast to Fig. 8, the horizontal slices through the gaps of the checkerboard pattern where the test model perturbations vanish are shown in the

left column while the slices through the extrema of the test model perturbations are shown in the right column. At $130\,\mathrm{km}$ depth (Fig. 9a) where the test model is unperturbed, the inversion produces a velocity perturbation very similar to the one at $175\,\mathrm{km}$ depth indicating severe vertical smearing. This behaviour is due to the subvertical ray paths at shallow levels in the upper mantle that intersect at an acute angle leading to loss of vertical resolution. We estimate that the checkerboard anomalies are smeared vertically by at least $20\,\mathrm{km}$ at shallow depths below the crust (i.e. up to $\sim 150\,\mathrm{km}$ depth). This smearing effect

highlights the importance of an accurate model of the crust and also the uppermost parts of the mantle from independent data.

At greater depths, e.g. in the following layer slicing directly through the perturbed test model at a depth of $175\,\mathrm{km}$ (Fig. 9b), the checkerboard pattern is well recovered. However, amplitude recovery is worse compared to the coarser checkerboard test model and also vulnerability to NW/SE smearing is greater. In contrast to the uppermost unperturbed layer, the next one below at $220\,\mathrm{km}$ depth (Fig. 9c) is recovered much better due to the less steep rays at these depths. Smearing artifacts are faint getting







**Figure 8.** Results of checkerboard analysis after 12 iterations for the $3 \times 3 \times 4$ grid. Misfit reduction is 87 %. Horizontal slices in the left panel cut through the maxima of velocity anomalies. Slices in the right column cut through the unperturbed spaces in between.





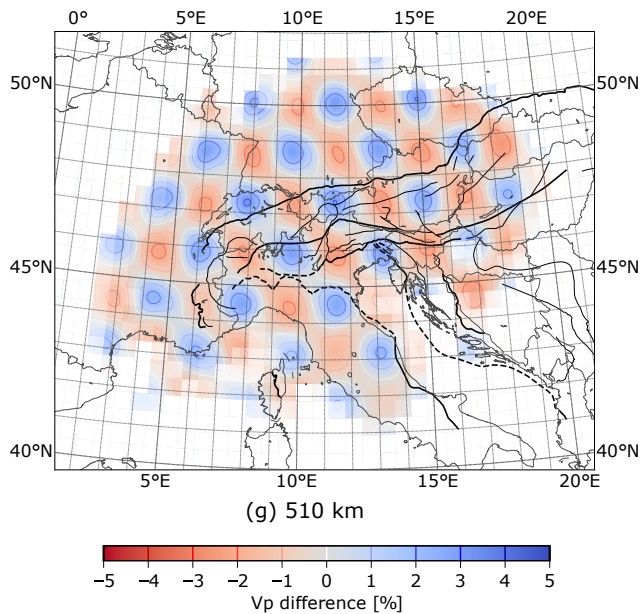

(g) 510 km

Vp difference [%]

**Figure 8.** Continued.

stronger close to the boundaries of the well irradiated model domain. The following perturbed layer at 265 km depth (Fig. 9d) is again recovered well but smearing of anomalies in the NW/SE direction connecting velocity perturbations of the same sign is more prominent. Below this depth, pattern recovery worsens systematically with depth. At 310 km (Fig. 9e), the unperturbed layer of the test model can still be distinguished from the perturbed ones but exhibits a clear imprint of the surrounding perturbed layers caused by smearing. Finally, for depths greater than 400 km (Fig. 9g-j) we can no longer distinguish between perturbed and unperturbed layers of the test model. Hence, we loose depth resolution for objects smaller than $\sim 50$ km in size at those depths.

Vertical slices through the checkerboard models (Fig. 10) further illustrate vertical and lateral smearing as discussed above. There is a slight vertical elongation of the reconstructed anomalies of the checkerboard in the uppermost level at $\approx 100$ km depth indicating a reduction of resolution below crustal depths owing to sub-vertical ray paths. For both, the coarse and the fine checkerboard, the separation of positive and negative anomalies is better reconstructed at middle depth levels (200 to 400 km). There is also some diagonal smearing which is weak in the central regions but strong at the model boundaries reflecting the fact that teleseismic rays propagate obliquely through the model domain. If there is insufficient ray crossing structures tend to become elongated parallel to the ray direction. This effect is best visible at the lateral model boundaries where crossing rays entering from the opposite side are missing but may also appear elsewhere if illumination from either side is strongly unbalanced.

## 5   Results





**Figure 9.** Results of checkerboard analysis after 12 iterations for the $2 \times 2 \times 3$ grid. Misfit reduction is $72\%$. Here, slices in the right panel cut through centre of velocity anomalies. Left panel slices cut through unperturbed space in between. Location of anomalies in the input model is highlighted by contour lines of $5\%$ velocity change in positive (dotted lines) and negative (dashed lines) direction.

## 5.1 Setting global damping and smoothing weights

Starting from the a priori model of the crust and the uppermost mantle and the laterally homogeneous model of Fig. 4 (denoted by $m_0$ in eq. (5)), we invert the dataset of traveltime residuals for P-wave velocity in the entire model domain. A priori





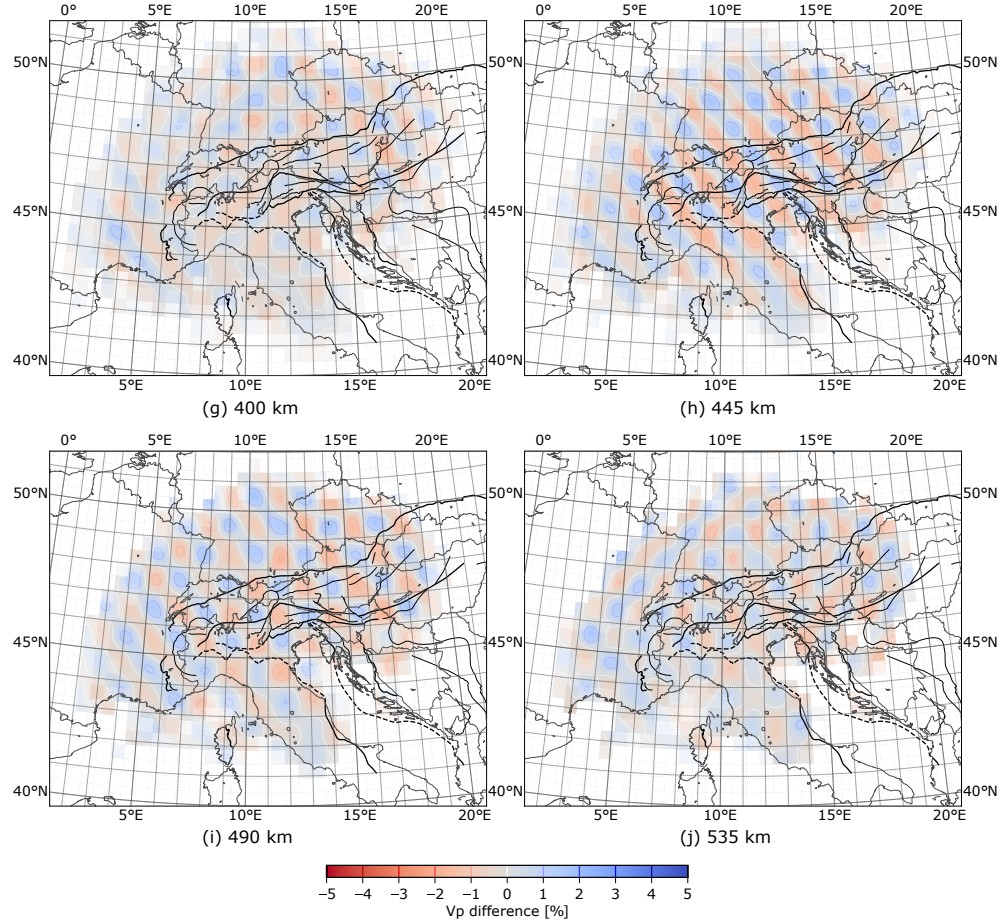

**Figure 9.** Continued.

standard deviations of P-wave velocity on the grid points are chosen as described in section 2.4 for the crust and uppermost
mantle. Below we choose a value of 15 % of the absolute velocity of the 1D background model. As resolution diminishes with
depth, smoothing is increased with depth by weighting the rows of the matrix $D$ in eq. (5) with a linearly increasing factor
between 1 (top) and $\sqrt{2}$ (bottom of the model). For setting the global damping and smoothing weights as introduced in eq. (5),
we carried out a series of inversions with varying values of these quantities. The result are so-called trade-off curves that reflect
the competing behaviour of misfit and model norm. We considered values for the damping weight of 3, 10, 30 and 100. Values
of the global smoothing weight comprise 10, 18, 30, 56 and 100. For each combination of these two parameters, we obtain a
value of the misfit and the model norm after 12 iterations. The resulting curves (Fig. 11) confirm the expectation that strong
regularization leads to high misfit and low norm and vice versa. The ideal values of the regularization weights are those that
provide the best compromise between the conflicting aims of low misfit and low model norm. Based on the results in Fig. 11,
we choose the values of 10 for the damping and 30 for the smoothing weight.


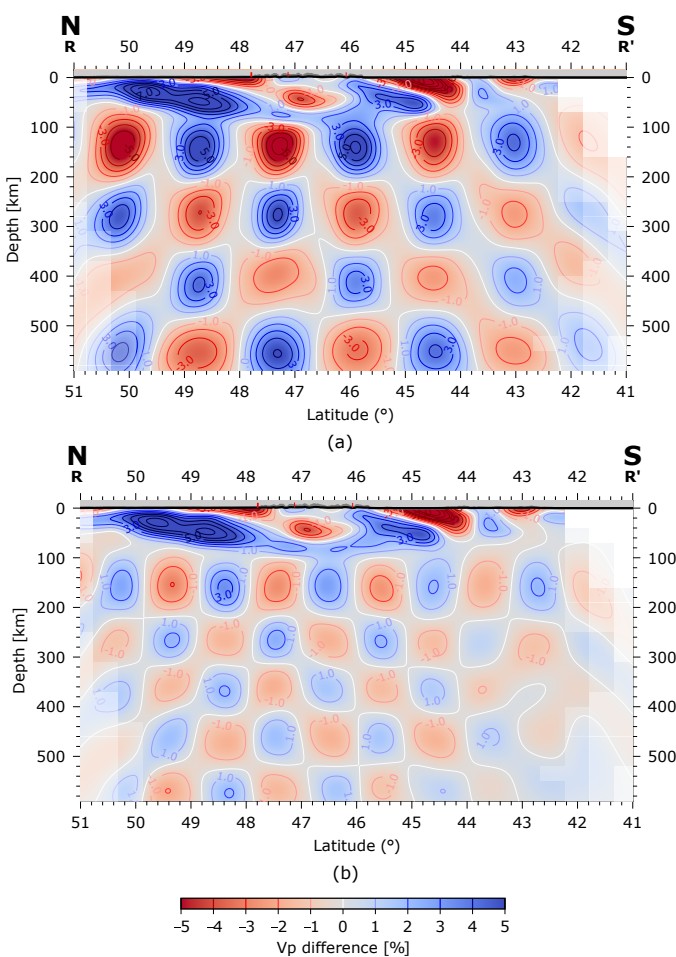

**Figure 10.** Vertical slices of profile R (Fig. 7) cutting through the checkerboard models at $11.5°$E.

In principle, we want to reduce the average misfit $\chi^2/N$ to 1 in order to explain the data within their uncertainties. This criterion could be used to regulate the number of iterations to be done. However, our estimates of uncertainty might be too conservative or too optimistic. Moreover, the misfit is not only controlled by the data uncertainties but also by other factors. First, our solution of the forward problem has errors; second, ray theory is only an approximation to the full wave propagation problem and, third, our model discretization does not admit all possible variations of P-wave velocity that may exist in the earth (including the ones outside the model domain). Hence, we expect additional deviations of the predictions from our measurements than the ones caused by the data uncertainty. As a consequence, a new criterion when to terminate the iterative inversion is needed. Motivated by the basic aim of low misfit and low norm, we monitored both quantities with the iterations (Fig. 12) and find that the misfit drops quickly after the first few iterations and then decreases very slowly. The total norm also rises quickly and then levels off but continues increasing steadily with iterations. For the models shown in this section, we choose an upper limit of 12 iterations because beyond that we pay for additional marginal misfit reduction by an over-





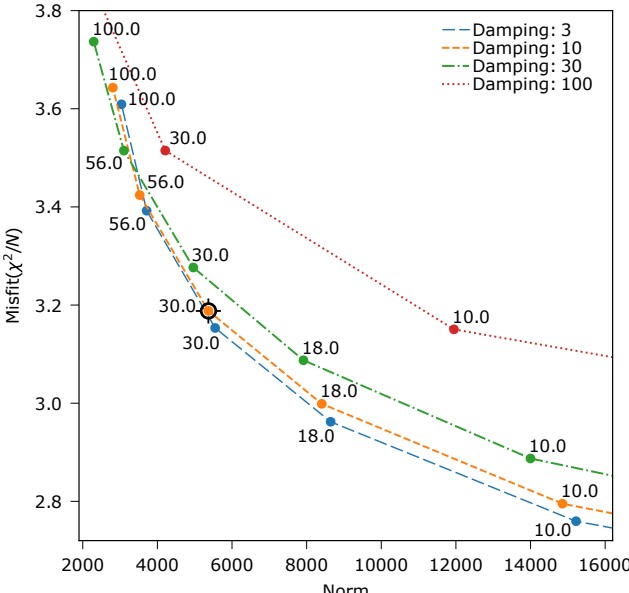

**Figure 11.** Data misfit against model norm for different smoothing and damping parameters. Different damping parameters indicated by line and point colors. Smoothing parameters are written as text besides data points in the plot.

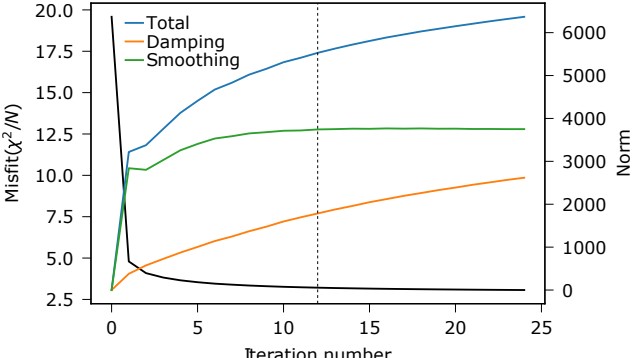

**Figure 12.** Misfit reduction for 24 inversion iterations. Smoothing and damping norm are displayed by the green and orange lines, total norm is displayed in blue. We use the model after iteration 12 as our final model, as the misfit reduction becomes insignificant hereafter, whilst the smoothing term (model roughness) has reached its maximum and only the damping norm (model variation) continues to grow.

proportional increase of model norm. In addition, the smoothing norm saturates after 12 iterations. This fact indicates that further iterations only increase the amplitude but only slightly the shape of the velocity perturbations.





## 5.2 Visualization of the velocity model

In the following, we show horizontal and vertical cross sections through the final velocity model. The figures show velocity perturbations relative to the 1D reference earth model (Fig. 4) including the velocity perturbations in the crust and uppermost mantle which are mainly controlled by our a priori crustal model.

In all cross sections, we overlay the velocity perturbations by a white shading that reduces intensity of the displayed colors. The amount of shading is chosen based on the illumination cells in the size of $2 \times 2 \times 2$ grid points by intersecting rays. To
measure intersection, we define four azimuthal quadrants and count the number of rays hitting the cell from each quadrant. If there are 5 rays from each quadrant, the illumination quality of the cell is set to 1 and no shading is applied. For each quadrant with less than five rays, the quality factor is reduced by 0.25 down to 0 if all quadrants have less than 5 rays and shading is strengthened accordingly. In this way, the information about illumination and ray crossing in each cell of the inversion grid is conveyed to the model cross sections.

## 5.3 Horizontal Cross Sections


The horizontal cross sections (Fig. 13) show velocity perturbations starting at a depth of 90 km and proceeding in 30 km and 50 km steps down to 500 km depth. The most conspicuous structures in the model are the high-velocity anomalies appearing in the depth range from 150 km to 210 km. They follow the Alpine chain and separate into three segments; a western one (C1), a central one (C2) and an eastern one (E). In addition, there is a fourth anomaly below the Apennines (A) that can be traced to
the southeast into central Italy. Moreover, we define two other anomalies best visible at 90 km depth, (W) close to the Penninic Front and (D) in the northern Dinarides.

At 180 km depth, the fast anomaly (A) extends from the southern-central Po plain in the north (10°E, 45°N) along the chain of the Apennines to central Italy (14°E, 42°N). At its northern end it is connected to anomaly (C1). Towards shallower depths from 150 km to 90 km, it successively migrates to the west and looses amplitude in particular in its southern parts. At these
depths, there is no connection to anomaly (C1) anymore. Below 180 km anomaly (A) keeps its lateral position down to 240 km depth but then moves successively to the west from 270 km to 400 km depth. It splits and fades out below 400 km.

The anomaly (C1) which appears at 180 km depth beneath the westernmost part of the Po plain migrates roughly in a NW-direction towards shallower depths indicating a SE dip in this shallow part. At 90 km depth it is still recognizable but its velocity perturbation is about half of the one at 180 km depth. Below 180 km this anomaly slightly migrates to the south/southeast until
about 300 km depth and develops a link to anomaly (C2) towards the northeast. Its amplitude decreases continuously with depth to finally vanish below 350 km.

Anomaly (C2) has its center at 10.5°E and 46.5°N at 180 km. It becomes rather weak at 150 km depth and is replaced by a strong low-velocity anomaly at 120 km. At 90 km the velocity perturbation at the same position is still negative but weaker than at 120 km. The positive anomaly at 47.5°N and 9.5°E slightly to the NW of this location may be structurally associated
to anomaly (C2). Between 180 km and 350 km, anomaly (C2) slightly migrates towards south/southwest, joins with anomaly (C1) to the southwest and becomes indiscernible at 400 km.





**Figure 13.** Horizontal slices through the resulting model. Traces of vertical profiles are indicated at 180 km depth (and as thin grey lines on all profiles) where the major velocity anomalies can be seen best. Important anomalies discussed in the text are marked at 90 km and 180 km depth.







**Figure 13.** Continued

Anomaly (E) is centered at 13°E and 47°N at 180 km and stretches over three degrees in longitude. It rapidly weakens at 150 km and is replaced by low velocities at 120 km depth. There is a small patch of elevated velocities to the west of the original



location at 120 and 90 km depth which may be structurally related to anomaly (E). From 180 km to 240 km depth, the anomaly
keeps its intensity and exhibits a very marginal shift of its center towards east. At 270 and 300 km depth, it seems to split into
a northern and a southern part that merge again at 350 km. The anomaly becomes indiscernible below 400 km.

In contrast to the strong high-velocity anomaly (A) beneath the Apennines, there is no comparable region of high-velocities
at the eastern rim of Adria beneath the Dinarides. Velocities there are normal to slightly positive except at 120 and 90 km depth
where we obtain elevated velocities beneath the northermost Dinarides (D) and also further to the SE.

Negative velocity perturbations generally surround the described high velocity anomalies to the north, west and east in
the depth range from 120 km to 270 km. Low velocities also prevail in the same depth range to the south and southeast of
the high-velocity anomalies stretching from the Po plain into Adria. Exemptions from this general pattern is the strong low-
velocity anomaly at 120 km depth located at 10.5°E and 46.5°N that separates the fast anomaly (C2) from shallower levels, and
the extension of the low-velocity anomaly beneath the southeastern Po plain to the NW, also at 120 km depth, that separates
anomalies (A) and (C1). The low velocity regions to the northwest of the fast anomalies (C1), (C2) and (E) tend to migrate to
the southeast with depth. They are strongest at 90 and 120 km depth where they are located beneath the Black Forest in SW
Germany and the Swiss Jura.

The described regular pattern of fast and slow anomalies dissolves at 350 km depth. Below, we obtain patchy but generally
positive velocity perturbations beneath the entire Alpine region stretching from southern France to the Pannonian Basin. Low-
velocities are mostly restricted to the northern foreland and central Italy.

## 5.4 Vertical Cross Sections

We display three vertical sections (Fig. 14) through the resulting model that cut centrally through the major positive velocity
anomalies in the Alps and Apennines at 180 km depth (Fig. 14). Moreover, profile P2 passes through the conspicuous low-
velocity anomalies in the NW-Alpine foreland at depths above 180 km and, of course, through the one directly located on top
of fast anomaly (C2). All three profiles cross the strong low-velocity anomaly underneath the Po plain appearing from 90 to
about 270 km depth. The traces of the profiles are shown in Fig. 13d and as weak gray lines in all other figure panels.

Profile P1 (Fig. 14a) begins in SE-France, crosses the western Alpine mountain chain, the Po plain and the Adriatic Sea and
ends in the northern Dinarides. Its major feature is a high-velocity-anomaly reaching from crustal depths in the westernmost
part to depths of about 100 km close to the Penninic Front below the western margin of the Alpine mountain chain. Then,
the anomaly becomes horizontal, weakens slightly and starts to dip again more steeply beneath the Ivrea zone continuing in
upward curved banana shape towards the east. Above this feature at crustal to uppermost mantle levels, separated by a strong
low-velocity region, there are strong high velocity anomalies, one associated with the Ivrea zone and the other one with Adriatic
lithosphere and possibly the northern Dinarides. Below the banana-shaped fast anomaly, at 8 to 9°E, there is a rather weak,
steeply dipping high velocity anomaly which may reach down to 500 km depth. At the bottom west of the section, there is a
large, seemingly horizontal high-velocity anomaly.

Profile P2 (Fig. 14b) is NW to SE oriented and begins in the northwest at the western border of Germany, crossing the central
Alpine mountain chain and the Po plain until it reaches the Apennines and ends in southern Italy. It is directed along the main





**Figure 14.** Three vertical profiles cutting through the four major anomalies in the Alps and Apennines as indicated in Fig. 13d: (a) The western Alpine slab, (b) the central Alpine slab and (c) the eastern Alpine slab in the east, as well as the Apenninic slab in the west. Intersections between the different profiles are marked by vertical dashed lines.





smearing direction as discussed in Sect. 4. Therefore, it is important to be aware of possible stretching of anomalies parallel
to this profile. The main feature is the strong fast anomaly starting at crustal levels in the NW and dipping to the SE down to
about 120 km depth. Directly below, dipping parallely and with identical lateral extent, there is a rather strong low-velocity
zone. The fast anomaly is interrupted by an about 40 km thick low-velocity region and then continues with a slighty stronger
dip to about 360 km depth. To the SE above this anomaly, there is an extended low-velocity zone beneath the Po plain that was
already mentioned in context with profile P1. Above it, there is again a shallow high-velocity area associated with Adriatic
lithosphere. Even further to the SE the profile touches the extended high-velocity anomaly beneath the Apennines.

Profile 3 (Fig. 14c) begins in southern Poland in the northeast, crosses the eastern Alps and the eastern Alpine high velocity
anomaly as well as the Apennines almost perpendicular to their strike and ends in Corsica. It demonstrates that the strong
positive anomaly (E) appearing below 150 km is neither connected to European lithosphere in the NE (between 50 and 48°N)
nor to Adriatic lithosphere in the SW between 46 and 44.5°N. Its southwestern extension more likely belongs to anomaly (C)
defined in profile P2. There is a potential vertical continuation of anomaly (E) that splits into a part dipping SW and another
stronger part dipping NE. Further to the SW, the profile intersects the high-velocity anomaly beneath the Apennines (A) that is
well separated by a region of very low velocities from shallow Adriatic lithosphere further to the NE. Anomaly (A) is connected
to the crust, dips nearly vertically with a slight tendency towards the NE in the shallow mantle and then overturns towards SW
below 200 km.

## 6    Discussion

We follow the tradition of previous authors (Lippitsch et al., 2003; Mitterbauer et al., 2011; Piromallo and Morelli, 2003;
Spakman et al., 1993) and interpret the high-velocity anomalies as associated with lithospheric material subducted during the
different stages of the collision of Europe and Adria. The fast anomalies defined in the horizontal and vertical sections above
correspond to the Adriatic slab beneath the Apennines (A), a central, arcuate slab following the western end of the Periadriatic
Fault System (PFS) of presumably European origin (C1, C2), and a slab in the Eastern Alps (E) whose provenance is currently
debated due to conflicting images of its dip (Lippitsch et al., 2003; Mitterbauer et al., 2011) leading to theories of a possible
slab polarity switch in this area  (Kissling et al., 2006; Lippitsch et al., 2003; Schmid et al., 2004; Luth et al., 2013). These
anomalies are clearly visible as rather well-separated entities in our model in the depth range from 150 to 240 km but loose
their individual identity at both smaller and greater depths. Hence, attributing these anomalies to lithospheric slabs of European
or Adriatic provenance is subject to interpretation and should also be based on additional evidence. The shallow fast anomaly
(W) close to the Penninic Front is interpreted as the Western Alpine slab. Anomaly (D) indicates shallow subducted material
beneath the northern Dinarides. The wide-spread high-velocity anomalies at the bottom of the model underlying the entire
Alpine chain are presumably relics of the Alpine Tethyan ocean.

Low velocity anomalies are generally attributed to warm asthenospheric mantle. However, as discussed in detail in a com-
panion paper by Handy et al. (2021), the distribution of low-velocities in our model (horizontal sections at 120 km to 210 km
depth, Fig. 13b-e) and Fig. 14b) indicates a dichotomy in the character of low velocities in the western and eastern Alpine





region. Handy et al. (2021) postulate that the low-velocity volumes extending into the western and northwestern foreland represent mantle material that is inherited from pre-Alpine compositional domains. It is attached to the base of the dipping slabs and exhibits the same kinematic behaviour. Handy et al. (2021) use the term European tectosphere for these structures. On the other hand, low velocities below the eastern Alps and the Carpathians are interpreted as upwelling asthenospheric mate-

rial associated with a large-scale delamination of European lithosphere and orogen-parallel eastward extrusion of the eastern Alpine and Carpathian crustal edifice presumably driven by the rollback of the Carpathian slab. Also, the low-velocity region below the Po plain presumably represents asthenospheric upwelling combined with a contribution due to hydration related to the backarc position of this anomaly behind the descending European tectosphere (Giacomuzzi et al., 2011).

One issue regarding the interpretation of the fast anomalies concerns the depth extent and continuity of the slab beneath the

western Alps. Based on the vertical section along profile P1 (Fig. 14a), we argue for a relatively short (130 km), shallow western Alpine slab derived from the European lithosphere ending at about 100 km depth close to the Penninic Front. A slab length of about 130 km fits well with the W-E-component of the displacement path taken by the most internal part of the European crust in this area (the present day Gran Paradiso massif) with respect to stable Europe (Schmid et al., 2017). The banana-shaped fast anomaly following further east more likely represents the central Alpine slab which is cut by profile P1 obliquely to its

roughly SE-ward dip direction. The steeply dipping areas of elevated velocities beneath the banana-shaped anomaly are weak and we refrain from interpreting them as a possible continuation of the western Alpine slab. With this interpretation, we deviate from Zhao et al. (2016) who interpret this deep-reaching fast anomaly as part of the Western Alpine slab. It is more likely that the scattered areas of positive velocity anomalies at great depths represent relics of the Alpine Tethys. This interpretation is supported by the patchy but widespread fast anomalies covering the entire Alpine region at depths below 400 km. We also

deviate from the interpretation by Kästle et al. (2020) who interpret the banana-shaped anomaly as western Alpine slab and postulate a break-off at around 100 km depth.

The interpretation of a short western Alpine slab implies that the fast anomaly (C1) appearing from 150 to 240 km depth structurally belongs to the central Alpine slab which comprises the fast, arc-shaped anomalies following the Periadritic fault system from about 8°E to 10.5°E visible from 90 to 300 km depth (Fig. 13). Its easternmost part is featured in profile P2

(Fig. 14b). In this profile, the slab has a gap at about 120 km and continues below 160 km down to about 400 km with a steep SE-ward dip. Below the shallow part of the slab, we find a low-velocity zone following the slab dip and interpreted by Handy et al. (2021) as lower part of European tectosphere comprising both high and low-velocity anomalies. Detachments further to the west of the profile are not as clear as the slab anomaly is not intersected by low-velocity areas but only weakens laterally and seems to split into two vertical arms below 150 km depth in order to merge again below 240 km depth.

In contrast to the central Alpine slab, the eastern Alpine slab associated with the easternmost fast anomaly (E) at 180 km depth (Fig. 13) is completely detached and replaced by warm mantle above 150 km depth. The slab featured in profile P3 is very prominent between 200 and 300 km depth but weakens at greater depths and seems to split into two branches, a stronger northern and a weaker southern one. This behaviour may produce the overall impression that anomaly (E) slightly dips towards NE. We have carried out a dedicated resolution test to check if the detachment of the slab can be resolved by our data (Fig. 15).

For this purpose, we created test models containing an elongated, box shaped low-velocity perturbation of 3% underlain by a



fast, also box-shaped anomaly of up to $5\%$. In test models (a) and (c), the fast box dips towards S, in test model (b) towards north. The low-velocity box in test models (a) and (b) completely overlies the fast anomaly while in test model (c) the fast anomaly reaches up to the crust. Beside the test structures, the models also contain the velocity perturbations of our a priori crustal model. From the test models, we computed synthetic traveltime residuals, added Gaussian noise and inverted them using

the same parameters as in the original inversion. In all cases we notice a general decrease of the anomaly amplitudes due to regularization of the inversion. In the case of the detached slabs (Fig. 15d,e) the shape of the anomalies can be reconstructed well. However, the reconstructed detached slabs exhibit a stronger dip than the test structures especially for the N-dipping slab. Reconstruction is best for the S-dipping attached slab. We conclude from this result that we can resolve the detachment but may miss a weak northward dip of the slab.

Regarding the provenance of the eastern Alpine slab one might be tempted to postulate a structural connection to the Adriatic lithosphere in the SW instead of one to the European lithosphere in the NE. This cannot be decided from this tomography alone. However, Handy et al. (2021) argue for a European provenance of the slab as its down-dip length (300-500 km) matches the estimated Tertiary shortening in the Eastern Alps accommodated by south-dipping subduction of European tectosphere.

The largest high-velocity anomaly in our model is the Adriatic slab beneath the Apennines that is cut perpendicularly by
profile P3 (Fig. 14c). There, the slab exhibits a nearly vertical to slightly NE directed dip down to about 140 km and then turns towards SW. The latter dip direction is expected from plate tectonic reconstructions (Jolivet and Facenna, 2000) that describe a retreat of the Adriatic slab towards the east. The steepening of the slab at shallow depths may be related to the termination of the advancement of the Apennines in Plio-Pleistocene time (Handy et al., 2021). The Adriatic slab attains its greatest lateral extent below 180 km depth where it reaches well into central Italy (Fig. 13) reaching the southern limit of our model close to
$41°$N. Towards shallower depths, its SE-ward limit is systematically shifted towards NW. At 90 km depth its SE end is north of $43°$N. This behaviour indicates a detachment of the slab south of $43°$N. Full attachment of the slab to the crust seems to be restricted to its northernmost parts, being a potential reason for the overturning of the slab in its northern part because it inhibits or at least impedes a further eastward retreat of the slab. Remarkably, although having exhibited a dip in opposite directions in the Miocene (Jolivet and Facenna, 2000), the Adriatic slab and the SW part of the central Alpine slab merge at depths greater
than 180 km where they exhibit a nearly vertical dip. However, both are clearly separated partially by low-velocity regions at shallower depths.

With regard to the Dinarides, our model indicates a shallow slab only in their northernmost part. The high-velocity signature below 150 km along the Dinaridic chain is very weak and vanishes below 240 km. It should however be noted that resolution degrades in this part of model. In addition, the a priori crustal model is less reliable there than in the core region of the Alps.

The results of our study can be directly compared to the previous studies by Lippitsch et al. (2003), Mitterbauer et al. (2011) and Zhao et al. (2016) who investigated about the same region with data available before the installation of the AlpArray Seismic Network. In particular the study by Zhao et al. (2016) comprises the same region as covered in this study. The focus of the study by Mitterbauer et al. (2011) is in the eastern Alpine region while Lippitsch et al. (2003) concentrate on the core region of the Alps. All studies agree nearly perfectly in their velocity anomalies in the depth range around 200 km showing
the negative anomalies (C1), (C2) and (E) at the same locations. In the model by Lippitsch et al. (2003) and Mitterbauer et al.




(2011) (only anomalies (C2) and (E) because (C1) is outside the model domain), these anomalies appear well separated as in our study while they seem laterally connected in the model of Zhao et al. (2016).

All models diverge from each other in different ways towards shallower depths. In Lippitsch's model, both anomalies (C2) and (E) are continuous up to 90 km depth while anomaly (C1) disappears at 120 km and reappears at 90 km. Especially, the
upward continuity of anomaly (E) and its connection with fast Adriatic lithosphere to the SW lead to the interpretation of a NE-dipping slab of Adriatic provenance, also implying a polarity switch from the SE-dipping central Alpine slab to the NE dipping eastern Alpine slab. In contrast, in Mitterbauer's model anomaly (E) weakens with decreasing depth and appears to keep its lateral position indicating a rather vertical slab with weak or no connection to the crust and therefore no longer requiring a polarity switch. Zhao's model also shows a nearly vertical eastern Alpine slab with weakening positive anomaly
towards shallower depth. Whether the slab connects to the crust cannot be discerned from the provided figures. In our model, the fast anomaly (E) completely disappears at depths less than 120 km indicating a complete detachment. The dip is subvertical and, hence, does not require a polarity switch. Given the big amount of subducted material, Handy et al. (2021) argue for a European provenance of the slab as southward crustal shortening in the southern Alps is only about 50 km. Thus, the more plausible interpretation is that there is only one European slab most of which is detached from the orogenic crust. With regard
to anomaly (C2), both Mitterbauer's and Lippitsch's model do not show the low-velocity gap that in our model interrupts the central Alpine slab in its easternmost part.

Models differ from each other also at depths below about 250 km. For example, the three strong anomalies (C1), (C2) and (E) start to weaken and disappear already at 270 km in Lippitsch's model while they prevail down to a least 350 km in our model. This leads to a higher estimate of the length and penetration depth of the slabs. Similarly, Mitterbauer's model looses track
of the (C2) anomaly below 250 km. Their eastern Alpine slab looks coherent to great depths while in our model it develops a complicated internal structure. Only Zhao's model exhibits similarities to our model also at greater depths, for example the high-velocity regions underneath the entire Alpine chain below about 400 km depth as well as the deep-reaching European and Adriatic slabs. Comparison at great depth generally becomes difficult owing to the different resolution of the models at these depths.
The fact that the various models exhibit significant discrepancies particularly at shallow depths raises the question which role the crustal corrections play in this. All models differ in the crustal models available for correction and how the crustal corrections are applied. The previous studies directly apply the corrections to the observed traveltimes either assuming vertical wave propagation through the crust or calculating traveltime corrections for each event separately, whereas we integrate the a priori model into our initial velocity model and allow modifications during the inversion. Moreover, this study is the only one
that uses an a priori model of the crust and uppermost mantle that is based on a tomographic study using local earthquakes. Directly applied a priori crustal corrections may create compensating structures in the mantle below especially if they are in conflict with the observed teleseismic traveltimes. For example, a wrong low-velocity anomaly in the crust would produce a compensating high-velocity anomaly in the mantle below and vice versa. In this case, it is favourable if the teleseismic inversion is allowed to change the structure also in the crust.





A further effect that may add to discrepancies at shallow levels is the fact that in teleseismic tomographies vertical resolution tends to diminish towards shallow depths. It can be expected that especially tomographies based on smaller data sets may therefore be more sensitive to vertical or oblique smearing at shallow depths compared to others thereby worsening the tendency to image continuous vertical or dipping structures. Nevertheless, even with the data from the AlpArray Seismic Network we must rely on the a priori crustal model to be able to make robust statements about slab detachment and delamination.

Instead of providing a hand-drawn sketch of the 3-dimensional configuration of the slabs in the greater Alpine region as in Handy et al. (2021), Zhao et al. (2016) or Kästle et al. (2020), we display here a 3-dimensional figure directly created from our model showing the outer boundaries of volumes in the model whose velocity perturbation is greater or equal than 1.25% (Fig. 16). We have chosen a view from the SE that allows to visualize all the major slabs (except the Western Alpine one) discussed in this paper. High-velocity structures in the SE that would block the view have been eliminated. Many of the

features discussed above can be recognized in this figure, for example, the central European slab going down to 410 km depth (indicated by the thin yellow line) thereby decreasing its dip angle, bifurcating around 180 km depth (indicated by the thin brown line) and again merging at shallower depths; the detachment of parts of the central and the entire eastern Alpine slab; the overturned Adriatic slab detached in its southern parts but still attached further north and separated at shallow depths by a little hole from the European slab; the merging of the central Alpine and Adriatic slab at greater depths. The western Alpine

slab is not visible as it is hidden by the central one. The 3D-view demonstrates that our common perception of slabs as thin, regular sheets with planar or neatly curved surfaces is too simplistic and idealizing. Realistic slabs have an unregular shape, exhibit corrugated and torn surfaces with holes and gaps and may show strong variations in thickness.





**Figure 15.** Resolution analysis for profile E on longitude $13.3°$E. Top panel: 3D models of the three different test cases for (a) a delaminated slab dipping to the south, overlain by a low velocity-zone, (b) an identical case but with a northward dipping slab and (c) a slab connected to the European lithosphere with a low-velocity zone representing upwelling mantle. Meshed volumes show velocity perturbation iso-surfaces of P-wave velocity increase of $2\,\%$ of the actual model results. Semi-transparent rectangular boxes show positions of the modeled anomalies for the resolution test. Middle panel: Vertical profile through the resulting spatially smoothed synthetic slabs, after importing them into the velocity grid. Bottom panel: Recovered structures after 12 inversion iterations.





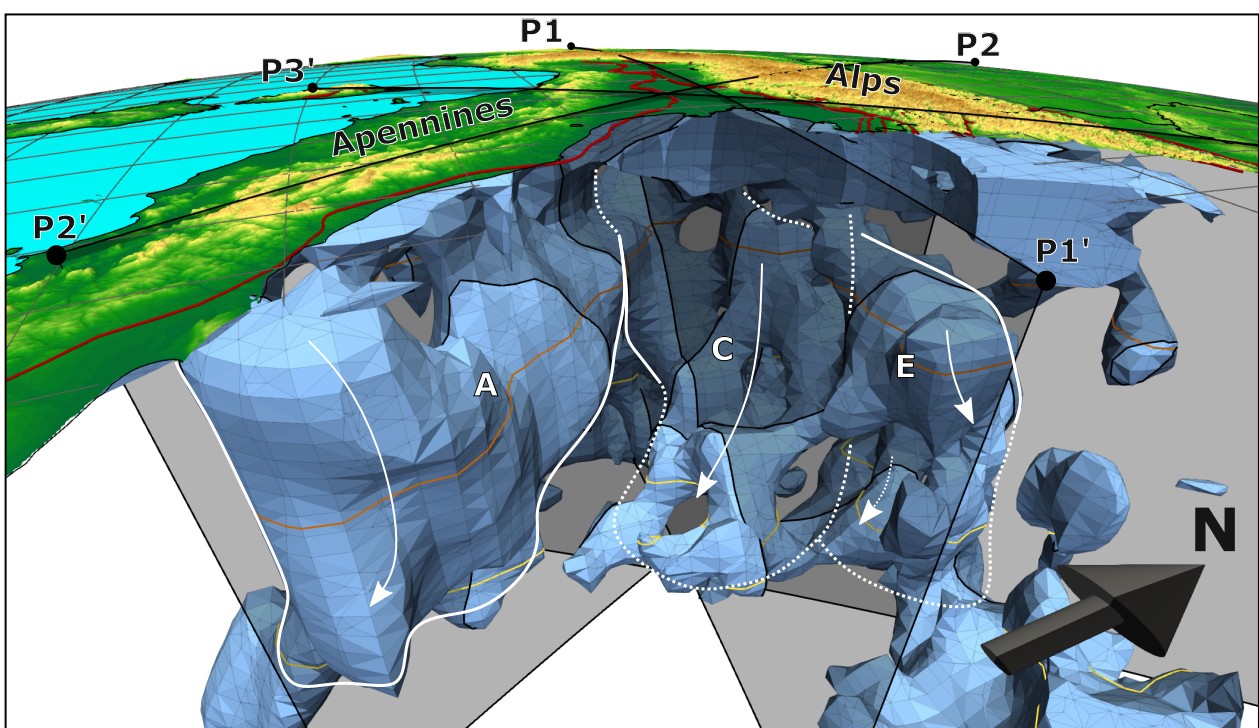

**Figure 16.** Three-dimensional view from the southeast (Adria) into the resulting model after 12 inversion iterations. We show here velocity iso-surfaces for an increase of $1.25\%$ in P-wave velocity. High-velocity structures in the SE that would block the view have been clipped. The three vertical profiles P1, P2 and P3 are indicated as semi-transparent surfaces as well as the respective contours of the iso-surfaces along these profiles (black lines). The $410\,\mathrm{km}$ discontinuity is shown as thin yellow line on the anomalies, as well as the outline of the horizontal slice at $180\,\mathrm{km}$ depth (thin brown line). Simplified shape and dipping direction of the different anomalies (A), (C) and (E) are indicated by white lines.



# 7 Conclusions

This study became only possible through the concerted effort of many geoscientists from various European countries in es-
tablishing the AlpArray Seismic network and sharing the recorded data with everybody who contributed to their acquisition.
The AlpArray network and dataset are unique in several aspects: AlpArray is the densest seismic network that ever covered
an entire orogenic belt including the forelands up to now, it was built from high-quality seismic broadband instruments and
was operated for an extraordinarily long time. Based on the wealth of waveforms recorded by AlpArray, we have been able
to extract an exceptionally large dataset of high-quality teleseismic traveltimes and traveltime residuals from which the new
exciting images of mantle structure beneath the greater Alpine region and the Apennines presented in this study are derived.

AlpArray waveforms proved to be particularly well suited to enhance traveltime determination by array methods such as
beamforming and cross-correlation required to fully exploit the resolution capabilities offered by such an array. Thanks to the
long recording time, it was also possible to obtain recordings from azimuthal sectors with very low teleseismic activity.

We have made particular efforts to compile the most recent and reliable information on P-wave velocity structure of the crust
and uppermost mantle in the study region which is essential to make inferences on the nature of coupling of the slabs with
the plates at the surface. In contrast to previous studies, the traveltime data were not directly corrected for crustal effects but
instead the a priori crustal model was incorporated as initial model into the teleseismic inversion. By defining a priori standard
deviations for the initial model, the inversion is regularized towards the a priori model but may deviate from it if demanded by
the data. In this way, we can avoid or reduce undesired compensation effects if the a priori model proves to be incorrect.

The new detailed images of slab configuration beneath the greater Alpine region and the Apennines exhibit differences to
previous studies in different places leading to a substantial change of the structural interpretation. The western Alpine slab
is short and ends already at about $100\,\mathrm{km}$ depth. Its apparent extension to the east and to greater depths interpreted by other
authors is rather due to the central Alpine slab that extends to the south at greater depths. The central Alpine slab follows
roughly the curved western end of the Periadriatic Fault System and bifurcates into two branches at depths between $150\,\mathrm{km}$
and $250\,\mathrm{km}$ depth. Contrary to previous work, it is detached at its eastern end by a horizontal slab tear at about $120\,\mathrm{km}$ depth.
There is a little gap between the central and eastern Alpine slab located beneath the western rim of the Tauern window. Also
in contrast to previous work, the eastern Alpine slab is completely detached from the orogenic root and develops its full extent
at around $180\,\mathrm{km}$ depth. The separation of central and eastern slab becomes unclear below $250\,\mathrm{km}$ depth. Both slabs dip
very steeply and exhibit connections to subducted material at the bottom of the model presumably representing relics of the
subducted Alpine Tethys. The new model does not require a slab polarity switch due to the detachment from the crust and the
nearly vertical dip of the eastern Alpine slab. Tectonic arguments rather suggest a European provenance of the slab. Thus, all
high-velocity zones imaged along the Alpine mountain belt represent subducted material of European origin. Large amounts
of subducted Adriatic lithosphere is found only beneath the Apenninic mountain range reaching great depths and showing a
slight overturning from NE to SW dipping mainly in its northern part where it is still attached to the crust. Further to the SW,
the Adriatic slab is detached by a horizontal tear that seems to deepen towards the SW.





The slow anomalies in the model exhibit an interesting spatial pattern that suggests a dichotomy of the properties of the upper mantle between the western and northern, and the eastern and southern regions of the study area. In the west and north, slow anomalies dip in parallel with fast anomalies hinting at a thick downgoing "tectosphere" of about $150-180$ km thickness that may be inherited from Variscan or pre-Variscan structures. In contrast, in the east and south, Adriatic lithosphere is comparably

thin and underlain by warm asthenosphere influenced in the south by a potential hydration of the mantle behind the European slab or, in the east, by eastward extrusion and thinning of the lithosphere owing to the retreat of the Carpathian slab.

Three-dimensional visualization of iso-surfaces of the high-velocity anomalies that reflect to some extent the "shape" of the slab make clear that subducted material in nature is rather different from the schematic cartoons of slabs drawn in textbooks and papers. Real slabs exhibit a complicated internal structure with holes, gaps, tears and bifurcations that reflects the spatial

variation of the velocity anomalies.





*Team list.* The AlpArray Seismic Network team: György HETÉNYI, Rafael ABREU, Ivo ALLEGRETTI, Maria-Theresia APOLONER, Coralie AUBERT, Simon BESANÇON, Maxime BÈS DE BERC, Götz BOKELMANN, Didier BRUNEL, Marco CAPELLO, Martina ČARMAN, Adriano CAVALIERE, Jérôme CHÈZE, Claudio CHIARABBA, John CLINTON, Glenn COUGOULAT, Wayne C. CRAW-FORD, Luigia CRISTIANO, Tibor CZIFRA, Ezio D'ALEMA, Stefania DANESI, Romuald DANIEL, Anke DANNOWSKI, Iva DASO-
VIĆ, Anne DESCHAMPS, Jean-Xavier DESSA, Cécile DOUBRE, Sven EGDORF, ETHZ-SED Electronics Lab, Tomislav FIKET, Kasper FISCHER, Florian FUCHS, Sigward FUNKE, Domenico GIARDINI, Aladino GOVONI, Zoltán GRÁCZER, Gidera GRÖSCHL, Stefan HEIMERS, Ben HEIT, Davorka HERAK, Marijan HERAK, Johann HUBER, Dejan JARIĆ, Petr JEDLIČKA, Yan JIA, Hélène JUND, Edi KISSLING, Stefan KLINGEN, Bernhard KLOTZ, Petr KOLÍNSKÝ, Heidrun KOPP, Michael KORN, Josef KOTEK, Lothar KÜHNE, Krešo KUK, Dietrich LANGE, Jürgen LOOS, Sara LOVATI, Deny MALENGROS, Lucia MARGHERITI, Christophe MARON, Xavier MARTIN,
Marco MASSA, Francesco MAZZARINI, Thomas MEIER, Laurent MÉTRAL, Irene MOLINARI, Milena MORETTI, Anna NARDI, Jurij PAHOR, Anne PAUL, Catherine PÉQUEGNAT, Daniel PETERSEN, Damiano PESARESI, Davide PICCININI, Claudia PIROMALLO, Thomas PLENEFISCH, Jaroslava PLOMEROVÁ, Silvia PONDRELLI, Snježan PREVOLNIK, Roman RACINE, Marc RÉGNIER, Miriam REISS, Joachim RITTER, Georg RÜMPKER, Simone SALIMBENI, Marco SANTULIN, Werner SCHERER, Sven SCHIPPKUS, Detlef SCHULTE-KORTNACK, Vesna ŠIPKA, Stefano SOLARINO, Daniele SPALLAROSSA, Kathrin SPIEKER, Josip STIPČEVIĆ, Angelo
STROLLO, Bálint SÜLE, Gyöngyvér SZANYI, Eszter SZŰCS, Christine THOMAS, Martin THORWART, Frederik TILMANN, Stefan UEDING, Massimiliano VALLOCCHIA, Luděk VECSEY, René VOIGT, Joachim WASSERMANN, Zoltán WÉBER, Christian WEIDLE, Viktor WESZTERGOM, Gauthier WEYLAND, Stefan WIEMER, Felix WOLF, David WOLYNIEC, Thomas ZIEKE, Mladen ŽIVČIĆ, He-lena ŽLEBČÍKOVÁ. The AlpArray SWATH-D field team: Luigia Cristiano (Freie Universität Berlin, Helmholtz-Zentrum Potsdam Deutsches GeoForschungsZentrum (GFZ), Peter Pilz, Camilla Cattania, Francesco Maccaferri, Angelo Strollo, Günter Asch, Peter Wigger, James
Mechie, Karl Otto, Patricia Ritter, Djamil Al-Halbouni, Alexandra Mauerberger, Ariane Siebert, Leonard Grabow, Susanne Hemmleb, Xi-aohui Yuan, Thomas Zieke, Martin Haxter, Karl-Heinz Jaeckel, Christoph Sens-Schonfelder (GFZ), Michael Weber, Ludwig Kuhn, Florian Dorgerloh, Stefan Mauerberger, Jan Seidemann (Universität Potsdam), Frederik Tilmann, Rens Hofman (Freie Universität Berlin), Yan Jia, Nikolaus Horn, Helmut Hausmann, Stefan Weginger, Anton Vogelmann (Austria: Zentralanstalt für Meteorologie und Geodynamik (ZAMG)), Claudio Carraro, Corrado Morelli (Südtirol/Bozen: Amt für Geologie und Baustoffprüfung), Günther Walcher, Martin Pernter,
Markus Rauch (Civil Protection Bozen), Damiano Pesaresi, Giorgio Duri, Michele Bertoni, Paolo Fabris (Istituto Nazionale di Oceanografia e di Geofisica Sperimentale OGS (CRS Udine)), Andrea Franceschini, Mauro Zambotto, Luca Froner, Marco Garbin (also OGS) (Ufficio Studi Sismici e Geotecnici - Trento)

*Author contributions.* Wolfgang Friederich developed the initial idea of the project. Marcel Paffrath developed the code, compiled the dataset and ran the calculations. Marcel Paffrath and Wolfgang Friederich prepared the article.

*Competing interests.* The authors declare that they have no conflict of interest.





*Acknowledgements.* We greatly acknowledge the contributions of the AlpArray temporary network Z3 (Hetényi et al., 2018) and AlpArray Working Group (2015) making this work possible. We want to thank the Deutsche Forschungsgemeinschaft (DFG) for funding the work within the framework of DFG Priority Programme "Mountain Building Processes in Four Dimensions (MB-4D)" (SPP 2017). Many thanks to Tobias Diehl for providing us access to his crust and upper mantle tomography model.

We want to acknowledge all permanent and other temporary seismic networks used in this study: Malet et al. (2015); Department Of Earth And Environmental Sciences, Geophysical Observatory, University Of München (2001); CERN (2017); Swiss Seismological Service (SED) At ETH Zurich (1983); University Of Zagreb (2001); Institute Of Geophysics, Academy Of Sciences Of The Czech Republic (1973); RESIF (1995); Institut De Physique Du Globe De Paris (IPGP) and Ecole Et Observatoire Des Sciences De La Terre De Strasbourg (EOST) (1982); GEOFON Data Centre (1993); Federal Institute for Geosciences and Natural Resources (1976); University Of Genova (1967); Kövesligethy

Radó Seismological Observatory (Geodetic And Geophysical Institute, Research Centre For Astronomy And Earth Sciences, Hungarian Academy Of Sciences (MTA CSFK GGI KRSZO)) (1992); Albuquerque Seismological Laboratory (ASL)/USGS (1988); INGV Seismological Data Centre (1997); MedNet Project Partner Institutions (1988); French Landslide Observatory – Seismological Datacenter / RESIF (2006); OGS (Istituto Nazionale Di Oceanografia E Di Geofisica Sperimentale) And University Of Trieste (2002); ZAMG - Zentralanstalt Für Meterologie Und Geodynamik (1987); OGS (Istituto Nazionale Di Oceanografia E Di Geofisica Sperimentale) (2016); Polish Academy

of Sciences (PAN) Polskiej Akademii Nauk (1990); RESIF (2018); University Of Trieste (1993); ZAMG - Central Institute for Meteorology and Geodynamics (2006); ESI SAS (Earth Science Institute Of The Slovak Academy Of Sciences) (2004); Slovenian Environment Agency (2001); Geological Survey-Provincia Autonoma Di Trento (1981); Leipzig University (2001); Jena (2009); McKee et al. (2018); Guéguen et al. (2017); Deschamps and Beucler (2013).

The authors would also like to thank all members of the AlpArray Seismic Network team and the AlpArray SWATH-D field team,
mentioned in the team list above. As well as the members of the EASI field team: Jaroslava Plomerová, Helena Munzarová, Ludek Vecsey, Petr Jedlicka, Josef Kotek, Irene Bianchi, Maria-Theresia Apoloner, Florian Fuchs, Patrick Ott, Ehsan Qorbani, Katalin Gribovszki, Peter Kolinsky, Peter Jordakiev, Hans Huber, Stefano Solarino, Aladino Govoni, Simone Salimbeni, Lucia Margheriti, Adriano Cavaliere, John Clinton, Roman Racine, Sacha Barman, Robert Tanner, Pascal Graf, Laura Ermert, Anne Obermann, Stefan Hiemer, Meysam Rezaeifar, Edith Korger, Ludwig Auer, Korbinian Sager, György Hetényi, Irene Molinari, Marcus Herrmann, Saulé Zukauskaité, Paula Koelemeijer,
Sascha Winterberg. For more information on the team visit www.alparray.ethz.ch.

A special thanks to the authors of Matplotlib (Hunter, 2007), providing a powerful toolkit for scientific data and map visualization with the help of Basemap.

Last but not least we want to thank the seismology group of the Ruhr-Universität Bochum, which helped to improve the quality of this work by numerous discussions and contributions.



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
