# Peer review of "Imaging structure and geometry of slabs in the greater Alpine area -A P-wave travel-time tomography using AlpArray Seismic Network data"

_Solid Earth, 2021_

## Referee Comment (RC1)

**Review of manuscript se-2021-58**

Dear Editor,

The manuscript entitled "*Imaging structure and geometry of slabs in the greater Alpine area - A P-wave traveltime tomography using AlpArray Seismic Network data*" presents the first *P*-wave traveltime tomography derived from the entire Alparray experiment dataset. Given the importance of this experiment for the understanding of the Alps, this thing in itself would argue for a prompt publication of this work!

The manuscript is well written, the figures are nice and informative. The structure is ok but I think that the description of the datased should arrive before the methods. The discussion section would also deserve clearly defined subsections. I regret that the dataset and its processing is not fully detailed in this document, though I acknowledge that the processing of such a large dataset deserves a dedicated publication. I thus propose (below) to the authors to add some supplementary information to increase the self-consistency of the article. In particular, I would like that the authors describe the quality check performed on the seismograms.

The method used is classical (ray theory traveltime tomography) and generally well described. The problem of the crustal model is however addressed in a novel way by integrating a 3D model based on local earthquakes tomography as a starting model for the 80 first kilometres of the model.
The consequence of this approach (but I am not sure of it) however caused me some confusion. I have concerns about checkerbord test vertical cross-section shown in FIG. 10. The observed smearing in the first 80 km of the output solution is dramatic and, I think, prevent from any interpretation of this part of the final model. The authors state in the text that this is caused by the geometry of ray coverage (coarser in the lithosphere) but another resolution test in FIG. 15 seems to rule out this proposition (no smearing at all...). It seems to me that the use of a more or less fixed solution in the lithosphere can explain this discrepancy.

Maybe there is something I miss but, given the fact that some important geological/geodynamical implications lie in the upper-most 100 km of the model, I think that the authors must clarify this point.

Given the fact that this point can take some time to be adressed I propose a moderate to major revision of the manuscript. You will find below my detailed comments and questions sequentially organised (not ordered by priority). I would like the authors to address those points in a revised version of the manuscript.

Best regards

**Questions and comments**

1. l. 86: "the AASN constitutes a massive improvement of observational coverage." Can the authors quantify this improvement ?

2. l. 146: Add the reference just after "FMTOMO".

3. Given the fact that there are OBS at approximately 4 km beneath the sea level and stations at more than 2 km elevation I imagine that the topography is taken into account. Can the authors say few words about that ?

4. section 3: I would suggest to present the dataset before the methods.

5. section 3: I know that data selection and processing is detailed in an other article, but I think that the present manuscript has to be self-consistent. Can the authors gather in a single paragraph the selection

criterions the have used to request the seismograms; for instance and for now, chosen minimum magnitude is indicated at l. 245 and epicentral distances range at l. 251. Can they indicate if the dataset has passed a quality check (and which one, in particular for OBS), as I imagine that some Mw=5.5 quakes can have low signal to noise ratio ?

I also would like the authors to justify their choice of filtering (Butterworth lowpass I guess, poles ?) the seismograms at 0.5 Hz. Given the fact that the majority of the stations are temporary stations, I suspect long-period noise level to be quite high at some sites. A bandpass filtering would have thus appeared more suitable.

6. l. 260, FIG. 6: Use the term "mode" instead of "maximum" to define the value that appears the most in the distribution.

7. Can the authors (quickly) explain how they estimate the pick uncertainties.

8. FIG. 8: Can the authors add the colorbar for panels a-f ?

9. FIG. 9: I do not see the dashed and dotted contour lines indicated in the caption. Indicate or remove the solid and dashed curved that correspond to "faults". Also to be consistent with FIG. 8, it could be more confortable for the reader to place slices that cut through the center of the anomalies to the left panels and the one that cut through unperturbed zones on the right panels. Is it possible to add the colorbar on the first part of the FIG. 9 ?

10. FIG. 10: If I understand well, the upper part of the initial model is composed by a linear combination of two 3D models (Diehl et al. (2009) and Tesauro et al. (2008)) and a 1D model. The $\pm 5\%$ checkerboard anomalies are thus imposed to this 3D model. Am I ok ?

Again, if I understand well, the "crustal" model is designed to be possibly (slightly ? l. 215) modified during the inversion. I thus wonder if the smearing that we see in the resolution test is not mostly caused by this parametrization and not by the ray coverage.

What makes me suspicious is that the zone with the strongest smearing appears to be close to 80 km thick, *i.e.*, close to the 77.5 km thickness of the initial "crustal" model. It is also to note that despite the change in size of the anomalies, the recovered geometry is almost excactly the same above 100 km in both cases. Can the authors comment on this point ?

Also, the authors explain that the smearing is mostly due to the fact that ray paths are mostly vertical beneath the array. If this is the case I would have expected the smearing in the lithosphere to be mostly vertical, which is not the case. Can they add an E/W vertical cross-section to see if this pattern is also present in this direction ?

11. FIG. 13–14 and text: If the resolution test shown in FIG. 10 is indeed strongly affected by smearing above 100 km depth, I think that the slice at 90 km cannot be interpreted and should thus be removed.

This is particularly critical for the cross-sections in FIG. 14 as the presence the "W" and "D" anomalies are mainly visible above 100 km depth.

12. l. 440: It is, I think, reasonable to interpret high-velocity anomalies as lithospheric slabs on the base of petrophysical and geodynamical arguments but not because of "tradition"!

13. l. 453–463: Low velocity anomalies are often observed around subducting slabs. Faccenda and Capitanio (2013) also propose that such anomalies are caused by strong seismic anisotropy in the vicinity of the slab. The low velocity pattern in the Western Alps do follow the trend that can be seen in Barruol et al. (2011). This can be discussed by the authors.

14. FIG. 15 and l. 493: I am confused by the results shown in this figure. FIGURE 10 shows that in the upper-most part of the model, there is a strong smearing to the South along line R1-R1'. The profile in FIGURE 15 is, I think, line R2 (on FIG. 13) – E1-E1' is not defined in map view–, I.E. about 1° to the east of R1-R1'.

On FIGURE 15, anomalies in the top 80 km are fully recovered, without smearing and without a significant loss in amplitude. How can the authors explain this contradiction ?

**Typos**

1. l. 208, 216, 218: Add the years to "Diehl *et al.*" and "Tesauro *et al.*".

2. FIG. 15: Center and bottom panels are both noted (d), (e) and (f).

**References**

Barruol, G., M. Bonnin, H. Pedersen, G. H. R. Bokelmann, and C. Tiberi (2011). Belt-parallel mantle flow beneath a halted continental collision: The Western Alps. *Earth planet. Sci. Lett.*, **302** (3–4), pp. 429–438. DOI: 10.1016/j.epsl.2010.12.040.

Diehl, T., S. Husen, E. Kissling, and N. Deichmann (2009). High-resolution 3-D *P*-wave model of the Alpine crust. *Geophys. J. Int.*, **179** (2), pp. 1133–1147. DOI: 10.1111/j.1365-246X.2009.04331.x.

Faccenda, M. and F. A. Capitanio (2013). Seismic anisotropy around subduction zones: Insights from three-dimensional modeling of upper mantle deformation and SKS splitting calculation. *Geochemistry Geophysics Geosystem*, **14** (1), pp. 243–262. DOI: 10.1002/ggge.20055.

Tesauro, M., M. K. Kaban, and S. A. P. L. Cloetingh (2008). EuCRUST-07: A new reference model for the European crust. *Geophys. Res. Lett.s*, **35** (L05313). DOI: 10.1029/2007GL032244.

---

## Referee Comment (RC2)

Review of manuscript se-2021-58

Dear Editor,

The manuscript entitled *"**Imaging structure and geometry of slabs in the greater Alpine area - A P-wave traveltime tomography using AlpArray Seismic Network data** "* presents P -wave traveltime tomography using the data acquired from the temporary network of AlpArray and permanent broadband stations in the study area. The manuscript is well written and well organized. The network and the analysis of the data set are presented on a related paper in sufficient detail. I think the manuscript has the quality to be published in Solid Earth but at least a moderate revision would be necessary.

Below are the details of my comments.

The number of stations used are quite large providing spatial coherence among the stations over the network. On the other hand, the number of events are relatively low (331) and some are lower magnitude events. What is the criteria for using Mw5.5 as threshold magnitude ? How many stations recorded the Mw5.5 events ? There is no information on the S/N of the low magnitude events. If the low magnitude events are recorded by fewer stations, would that create any bias as the average of traveltimes of each event is removed traveltime residuals?

The methodology is explained well and the parameters for the inversion are chosen appropriately. The misfit in Figure 12 drops fast during the first 3-4 iterations while stay almost flat for the rest. But the authors prefer to use the results after 12 iterations for very small improvements on the misfit. It would be valuable to present the results after fast drop in misfit (3 iterations) and after 12 iterations. Is there any overfitting the data by increasing the number of iterations ?

The checkerboard tests are done but requires more, to present vertical and horizontal smearing in more detail. The authors state that the checkerboard anomalies are smeared at least 20 km at shallow depths below the crust. But they do not provide any information over rest of the domain. The initial checkerboard depth models should be provided together with the recovered patterns (Figure 10). Spike tests would be valuable to monitor smearing over the solution space.

The incorporation of the crustal model and upper mantle (<100km) into the inversion is a nice idea although more tests are needed to understand the influence of strong constraints on the velocity perturbations. The results after inversion should be compared to the initial model for depths < 100 km. Is there any bias on the transition from strongly constrained upper part (>100km) to unconstrained lower part ?

The authors compares their results to the previous teleseismic tomographic works. It appears that Zhao et al 2016 used lesser amount of data and attained similar resolutions. It would be nice if the authors display vertical cross sections of few profiles from the tomographic images of the previous works (e.g, Koulavov et al, 2009; Zhao et al 2016 ..) crossing the same structures.

The presentation of the 3D model in Figure 16 is the least satisfactory part of the paper. The Figure 16 does not make any impression neither as geology nor a velocity model. Depending on the level of velocity contours a different image with different size of low velocity holes and slab thicknesses would appear. It appears that some of the research questions the authors posed in the introduction such as "how thick and how long are the descending slabs ? " remained unanswered. A geology based velocity model derived from the tomography can be used as synthetic test to better constrain the slab thicknesses, the extent of the low velocity zones.

---

## Author Comment (AC1)

**Review of manuscript se-2021-58**

Dear Editor,

The manuscript entitled "Imaging structure and geometry of slabs in the greater Alpine area - A P-wave traveltime tomography using AlpArray Seismic Network data" presents the first P-wave traveltime tomography derived from the entire Alparray experiment dataset. Given the importance of this experiment for the understanding of the Alps, this thing in itself would argue for a prompt publication of this work!

Dear reviewer,

Thank you very much for your detailed review which increases the quality of our manuscript! We tried to response to each issue point by point in this document.

The manuscript is well written, the figures are nice and informative. The structure is ok but I think that the description of the dataset should arrive before the methods.

We prefer not to put the description of the dataset in front of the methods, as we think that the knowledge on how travel times and their errors are calculated (Sect. 2.1) should be provided before quantifying travel time residuals and uncertainties.

The discussion section would also deserve clearly defined subsections.

Okay, done.

I regret that the dataset and its processing is not fully detailed in this document, though I acknowledge that the processing of such a large dataset deserves a dedicated publication. I thus propose (below) to the authors to add some supplementary information to increase the self-consistency of the article. In particular, I would like that the authors describe the quality check performed on the seismograms.

We will add some information on the process of event selection and quality control of the calculated travel time residuals.

The method used is classical (ray theory traveltime tomography) and generally well described. The problem of the crustal model is however addressed in a novel way by integrating a 3D model based on local earthquakes tomography as a starting model for the 80 first kilometres of the model.
The consequence of this approach (but I am not sure of it) however caused me some confusion. I have concerns about checkerboard test vertical cross-section shown in FIG. 10. The observed smearing in the first 80 km of the output solution is dramatic and, I think, prevent from any interpretation of this part of the final model. The authors state in the text that this is caused by the geometry of ray coverage (coarser in the lithosphere) but another resolution test in FIG. 15 seems to rule out this proposition (no smearing at all...).
It seems to me that the use of a more or less fixed solution in the lithosphere can explain this discrepancy.

As stated in a short comment to this review already, we are aware that the process of integrating the crustal model as part of the starting model has caused some confusion and should be described more clearly especially with regard to the checkerboard test. There is **no checkerboard perturbation** applied in the first ~100 km of the model below the surface! The anomalies in the first 100 km depth in Fig. 10 are simply the (nearly unchanged) a priori crust + uppermost mantle model. We included the a priori crustal model into the test model to examine its effect on resolution in the mantle below, and to study how possible smearing of anomalies in the uppermost mantle may leak into the crustal domain.
We will extend Fig. 10 by a subfigure showing the (input) test model and one showing the anomalies appearing in the crustal domain in the output model after inversion. Moreover, we will modify the text in the revised manuscript to avoid further misunderstandings.

Maybe there is something I miss but, given the fact that some important geological/geodynamical implications lie in the upper-most 100 km of the model, I think that the authors must clarify this point.

See above

Given the fact that this point can take some time to be addressed I propose a moderate to major revision of the manuscript. You will find below my detailed comments and questions sequentially organised (not ordered by priority). I would like the authors to address those points in a revised version of the manuscript.

Best regards

**Questions and comments**

1. l. 86: "the AASN constitutes a massive improvement of observational coverage." Can the authors quantify this improvement ?
   Yes. Compared to the work by e.g., Lippitsch, our dataset has 40 times as many travel time measurements! We will add a statement in the manuscript.

2. l. 146: Add the reference just after "FMTOMO".
   Okay, done

3. Given the fact that there are OBS at approximately 4 km beneath the sea level and stations at more than 2 km elevation I imagine that the topography is taken into account. Can the authors say few words about that?
   The model grid is extended by 15 km above sea-level. The topography is considered implicitly by correctly positioning the receivers in the 3D model grid according to their elevation.

4. section 3: I would suggest to present the dataset before the methods.
   See above

5. section 3: I know that data selection and processing is detailed in an other article, but I think that the present manuscript has to be self-consistent. Can the authors gather in a single paragraph the selection criterions the have used to request the seismograms; for instance and for now, chosen minimum magnitude is indicated at l. 245 and epicentral distances range at l. 251. Can they indicate if the dataset has passed a quality check (and which one, in particular for OBS), as I imagine that some Mw=5.5 quakes can have low signal to noise ratio ? I also would like the authors to justify their choice of filtering (Butterworth lowpass I guess, poles ?) the seismograms at 0.5 Hz. Given the fact that the majority of the stations are temporary stations, I suspect long-period noise level to be quite high at some sites. A bandpass filtering would have thus appeared more suitable.
   Regarding quality check we will add a paragraph as stated above. We also note here that we did not clearly say that we use a Butterworth bandpass filter between 0.03 and 0.5 Hz to exclude long-period noise and will also add this in the revised manuscript.

6. l. 260, Fɪɢ. 6: Use the term "mode" instead of "maximum" to define the value that appears the most in the distribution.
   Okay, done

7. Can the authors (quickly) explain how they estimate the pick uncertainties.
   We explain this shortly at line 123 and below and by equation (4).

8. Fɪɢ. 8: Can the authors add the colorbar for panels a-f ?
   We will split the figure differently over the two pages and add a second colour bar. Adding one to the current figure would consume to much space vertically.

9. Fɪɢ. 9: I do not see the dashed and dotted contour lines indicated in the caption. Indicate or remove the solid and dashed curved that correspond to "faults". Also to be consistent with Fɪɢ. 8, it could be more comfortable for the reader to place slices that cut through the center of the anomalies to the left panels and the one that cut through unperturbed zones on the right panels. Is it possible to add the colorbar on the first part of the Fɪɢ. 9 ?

*The dashed and dotted contour lines were removed as they overloaded the figures. Text relict in caption was removed.*

*The problem with the slice depths is that in one case the checkerboard domain (below 100 km) starts with a perturbed layer and in the other one with an unperturbed one. Switching sides would lead to an irregular ordering of slices with depth which would be a lot more confusing.*

10. FIG. 10: If I understand well, the upper part of the initial model is composed by a linear combination of two 3D models (Diehl et al. (2009) and Tesauro et al. (2008)) and a 1D model. The +/-5% checkerboard anomalies are thus imposed to this 3D model. Am I ok ?

   *No, there is no checkerboard perturbation in the upper 100 km.*

   Again, if I understand well, the "crustal" model is designed to be possibly (slightly ? l. 215) modified during the inversion.

   *Yes. "Slightly" means that the variations in the crustal domain are forced to remain very small by means of the regularization term.*

   I thus wonder if the smearing that we see in the resolution test is not mostly caused by this parametrization and not by the ray coverage. What makes me suspicious is that the zone with the strongest smearing appears to be close to 80 km thick, i.e., close to the 77.5 km thickness of the initial "crustal" model. It is also to note that despite the change in size of the anomalies, the recovered geometry is almost exactly the same above 100 km in both cases. Can the authors comment on this point ? Also, the authors explain that the smearing is mostly due to the fact that ray paths are mostly vertical beneath the array. If this is the case I would have expected the smearing in the lithosphere to be mostly vertical, which is not the case. Can they add an E/W vertical cross-section to see if this pattern is also present in this direction?

   *There is no notable smearing in the crustal domain as there is no checkerboard pattern in the upper 100 km! As the reviewer observes correctly, the inversion almost exactly retains the a priori crustal model which was included into the input test model. See also the comments above.*

11. FIG. 13–14 and text: If the resolution test shown in FIG. 10 is indeed strongly affected by smearing above 100 km depth, I think that the slice at 90 km cannot be interpreted and should thus be removed. This is particularly critical for the cross-sections in FIG. 14 as the presence the "W" and "D" anomalies are mainly visible above 100 km depth.

   *There is no notable above 100 km because the inversion almost exactly retains the initial model there. The slice at 90km depth mostly reflects the (crust and) uppermost mantle model by T. Diehl. Also see above.*

12. l. 440: It is, I think, reasonable to interpret high-velocity anomalies as lithospheric slabs on the base of petrophysical and geodynamical arguments but not because of "tradition"!

   *That was just meant as a phrase, but we will change this!*

13. l. 453–463: Low velocity anomalies are often observed around subducting slabs. Faccenda and Capitanio (2013) also propose that such anomalies are caused by strong seismic anisotropy in the vicinity of the slab. The low velocity pattern in the Western Alps do follow the trend that can be seen in Barruol et al. (2011). This can be discussed by the authors.

   *Thank you for the suggestion. We also considered a possible effect of mantle anisotropy by horizontal mantle flow perpendicular to the slab dip direction which is indicated by e.g., SKS splitting. We will include this aspect in the discussion.*

14. FIG. 15 and l. 493: I am confused by the results shown in this figure. FIGURE 10 shows that in the upper-most part of the model, there is a strong smearing to the South along line R1-R1'. The profile in FIGURE 15 is, I think, line R2 (on FIG. 13) – E1-E1' is not defined in map view–, I.E. about 1_ to the east of R1-

R1'. On FIGURE 15, anomalies in the top 80 km are fully recovered, without smearing and without a significant loss in amplitude. How can the authors explain this contradiction ?

Thank you. The profile annotation will be corrected to R1-R1'. This synthetic test was performed in the same way as the checkerboard test. There is simply no perturbation applied in the uppermost ~100 km for the checkerboard test.

**Typos**

1. l. 208, 216, 218: Add the years to "Diehl et al." and "Tesauro et al.".
Okay

2. FIG. 15: Center and bottom panels are both noted (d), (e) and (f).
Okay

**References**

Barruol, G., M. Bonnin, H. Pedersen, G. H. R. Bokelmann, and C. Tiberi (2011). Belt-parallel mantle flow beneath a halted continental collision: The Western Alps. Earth planet. Sci. Lett., 302 (3–4), pp. 429–438. DOI: 10.1016/j.epsl.2010.12.040.

Diehl, T., S. Husen, E. Kissling, and N. Deichmann (2009). High-resolution 3-D P-wave model of the Alpine crust. Geophys. J. Int., 179 (2), pp. 1133–1147. DOI: 10.1111/j.1365- 246X.2009.04331.x.

Faccenda, M. and F. A. Capitanio (2013). Seismic anisotropy around subduction zones: Insights from threedimensional
modeling of upper mantle deformation and SKS splitting calculation. Geochemistry Geophysics Geosystem, 14 (1), pp. 243–262. DOI: 10.1002/ggge.20055.

Tesauro, M., M. K. Kaban, and S. A. P. L. Cloetingh (2008). EuCRUST-07: A new reference model for the European crust. Geophys. Res. Lett.s, 35 (L05313). DOI: 10.1029/2007GL032244.

---

## Author Comment (AC2)

Review of manuscript se-2021-58

Dear Editor,

The manuscript entitled "*Imaging structure and geometry of slabs in the greater Alpine area - A P-wave traveltime tomography using AlpArray Seismic Network data*" presents P -wave traveltime tomography using the data acquired from the temporary network of AlpArray and permanent broadband stations in the study area. The manuscript is well written and well organized. The network and the analysis of the data set are presented on a related paper in sufficient detail. I think the manuscript has the quality to be published in Solid Earth but at least a moderate revision would be necessary.

Dear reviewer,

Thank you very much for your detailed analysis of our manuscript. Your comments and suggestion will help us increase the quality of our work. In this document we try to respond to each point mentioned one by one.

Below are the details of my comments.

The number of stations used are quite large providing spatial coherence among the stations over the network. On the other hand, the number of events are relatively low (331) and some are lower magnitude events. What is the criteria for using Mw5.5 as threshold magnitude ? How many stations recorded the Mw5.5 events ? There is no information on the S/N of the low magnitude events. If the low magnitude events are recorded by fewer stations, would that create any bias as the average of traveltimes of each event is removed traveltime residuals?

We will write a new paragraph in which we describe the selection criteria for each event (and also for an onset on each station) in more detail. We chose a lower boundary of Mw5.5 because we experienced that the number of available picks for distances > 35° is not satisfactory anymore. However, one could try to lower the magnitude limit in a future work to possibly increase the number of events from the poorly covered azimuths especially when using waveform correlation for travel time determination.
We set a lower limit (100) for the number of onsets required for each event to avoid a bias of the array average. We rather experienced variations in the number of travel times per event related to the number of available stations due to delays in the deployment of the AlpArray stations.

The methodology is explained well and the parameters for the inversion are chosen appropriately. The misfit in Figure 12 drops fast during the first 3-4 iterations while stay almost flat for the rest. But the authors prefer to use the results after 12 iterations for very small improvements on the misfit. It would be valuable to present the results after fast drop in misfit (3 iterations) and after 12 iterations. Is there any overfitting the data by increasing the number of iterations ?

We chose to continue the inversion iterations until the roughness (smoothing norm) of the model saturates. This happens after 12 iterations. Thus, up to the $12^{th}$ iteration we can still get more detail into the model. After step 12 the smoothing norm stagnates and only the damping norm rises indicating that only the amplitude of the anomalies increases.

We also experienced the exact same behaviour in various synthetic tests, where the resulting model further approached the test model when continuing inversion iterations even if the misfit reduction was small.
There should not be any risk of overfitting of the data, as the final misfit of ~3.2 has not yet reached the limit of 1. Below this limit, we would start fitting the uncertainties of the travel time residuals into the model (overfitting).

The checkerboard tests are done but requires more, to present vertical and horizontal smearing in more detail. The authors state that the checkerboard anomalies are smeared at least 20 km at shallow depths below the crust. But they do not provide any information over rest of the domain. The initial checkerboard depth models should be provided together with the recovered patterns (Figure 10). Spike tests would be valuable to monitor smearing over the solution space.

We describe resolution capabilities of the model with depth when we evaluate the checkerboard results for each depth slice in section 4. We also show representative vertical slices though the recovered checkerboard (Fig. 10).
The used model is a combination of a classical checkerboard test and a spike test as there is unperturbed space in between the checkerboard tiles (which can be seen as kind of spikes). Therefore, we can monitor smearing effects quite well with this model and decided not to do additional spike tests.

The incorporation of the crustal model and upper mantle (<100km) into the inversion is a nice idea although more tests are needed to understand the influence of strong constraints on the velocity perturbations. The results after inversion should be compared to the initial model for depths < 100 km. Is there any bias on the transition from strongly constrained upper part (>100km) to unconstrained lower part ?

We will try to highlight the influences on the uppermost ~100 km when we update the vertical sections through the checkerboard to clarify the way we created the checkerboard model. There it also becomes visible in which way artifacts can occur within the weakly constrained parts of the crustal model.
We do not understand the last question regarding the bias on transition from strongly to unconstrained parts.

The authors compares their results to the previous teleseismic tomographic works. It appears that Zhao et al 2016 used lesser amount of data and attained similar resolutions. It would be nice if the authors display vertical cross sections of few profiles from the tomographic images of the previous works (e.g, Koulavov et al, 2009; Zhao et al 2016 ..) crossing the same structures.

We will add a comparison to other models that are digitally available in the same spatial domain.

The presentation of the 3D model in Figure 16 is the least satisfactory part of the paper. The Figure 16 does not make any impression neither as geology nor a velocity model. Depending on the level of velocity contours a different image with different size of low velocity holes and slab thicknesses would appear.

The main idea behind Fig. 16 is to demonstrate the complexity of the model results and to get an idea of the three-dimensionality of the different features. There is of course always some

subjectivity involved, as what we see in the figure depends on the threshold value of the shown iso-surfaces. On the other hand, we learned that there is also a strong bias involved when 2-D slices through a 3-D model (or slabs) are presented, as perceived features such as a dipping direction or size will always depend on the exact profile positions and a natural slab does never follow only one direction along a single (or multiple selected) profiles from start to end as an idealized one would do. We experienced that a repositioning of a profile by less than ~50 km at only one end may significantly change the perceived geometric structures.

We should maybe clarify in the manuscript that the decision for a threshold value for the iso-surfaces is subjective and that the 3-D geometry changes for different values. Still, one should keep in mind that when trying to interpret idealized, discrete slab "boundaries" from a seismic velocity model, these are also often based on interpreting isolines (or iso-surfaces).

It appears that some of the research questions the authors posed in the introduction such as "how thick and how long are the descending slabs ? " remained unanswered.

We discussed the penetration depth of different slabs (and with that their lengths). We will try to answer questions regarding slab thickness in the discussion as well. However, it is difficult to make more than educated guesses on the slab thickness, as there are no discrete boundaries of a subducted (heated up) slab anymore, the further it is subducted, as its thermal signature blurs with time. Also, the discussed smearing effects, make such guesses difficult.

A geology based velocity model derived from the tomography can be used as synthetic test to better constrain the slab thicknesses, the extent of the low velocity zones.

The 3-D structure we show here is directly derived from the data and only depends on the chosen value of the iso-surfaces. For a geological interpretation of the velocity model (positive and negative anomalies) we want to refer here to the paper of Handy et al. (2021).

References

Handy, M., Schmid, S., Paffrath, M., Friederich, W., and the AlpArray Working Group: European tectosphere and slabs beneath the greater Alpine area – Interpretation of mantle structure in the Alps-Apennines-Pannonian region from teleseismic Vp studies, Solid Earth Discuss. [preprint], https://doi.org/10.5194/se-2021-49, in review, 2021.

---

## Author Comment (AC3)

**Dear Anne Paul,**

Thank you for your comments and remarks.

We decided to add another vertical profile slicing through the checkerboard tiles from W to E at 44.5°N as a supplement (Fig. 1 in this document). We do not think that a third resolution profile around 13.3°E (N-S) will add much valuable information, as the densest station coverage around SWATH-D is already included in profile R1, whereas the influence of the EASI experiment on our dataset is minor (due to its end in 2015).

There might be a little misunderstanding, as the crustal model does not reach down to only 77.5 km, but up to 99 km depth (which is the bottom boundary of Tobias Diehl's model). The transition between the 1D background model of Diehl and AK135 is at 77.5 km.

You will see in the updated Fig. 10 (Fig. 2 in this document) that a 90 km slice also partly slices through the a priori model. We do not think that the vertical resolution of our model between 80-130 km is too bad to show a horizontal slice at that depth. As far as we understood, the reviewer(s) wrongly assumed that the a priori crustal perturbations are artifacts of the checkerboard in the crust and concluded a severe resolution problem in this area. We created another resolution test with checkerboard perturbations also in the crustal domain (and without the a priori crustal model) to investigate vertical smearing in the uppermost mantle and crustal domain that we will also add as a supplement to the manuscript (Fig. 3 and 4 in this document). In this test we do not see a severe smearing problem in the depth range between 80-130 km. Along the N-S profile we can even resolve structures in the uppermost 40 km, north of 46°N, whereas south of 46°N and along the W-E profile the crustal domain is dominated by vertical smearing. However, the resolution in the uppermost mantle is still satisfactory along both profiles.

Regarding the original test that included the crustal a priori model we only see that there is more vertical smearing possible in the depth range between 80-130 km than for e.g., between 150 and 300 km depth, but we also see that e.g., for the 2x2x3 grid points checkerboard (Fig. 10) there is only moderate smearing of the underlying checkerboard anomalies (starting at 140 km depth) up to 120 km depth and no smearing above 100 km depth. A depth slice at 90 km depth through this test model would not be contaminated by the checkerboard perturbations below. Hence, the true depth of the anomalies in a 90 km depth slice might be less well determined than e.g., for the slices between 150 km and 300 km, but still acceptable for a meaningful interpretation.

Best regards,

Marcel Paffrath and Wolfgang Friederich

---

## Editor Decision (ED1)

Dear authors,

The revised version of your manuscript takes into account the reviewers' comments as well as mine. I therefore consider that it does not need further review by the reviewers.
I have read the revised version carefully myself, and have noted a series of corrections that you should be applied before the manuscript can be accepted for publication. A large majority of these comments deal with typos, incorrect English writing, or shortcuts in references to publications. They are listed below. Please note that line numbers refer to the authors's tracked change version, and not to the revised version.

A more important comment deals with your discussion of the potential effect of seismic anisotropy on the results of your (isotropic) teleseismic travel-time tomography (lines 506-513). You write that "steeply incident teleseismic P-waves passing through these mantle regions tend to be slower than the isotropic average because they propagate in a plane perpendicular to the fast axis". This is wrong in most cases because, in first approximation, teleseismic P waves propagate in the vertical plane defined by the source-receiver back-azimuth which is generally not perpendicular to the fast-velocity direction of the SKS anisotropy. However, it is right that anisotropy related to subduction induces artificial low-velocity anomalies around slabs in seismic tomographies of subduction zones. This effect has been studied by Bezada et al. (2016, doi: 10.1002/2016GC006507). You should correct this paragraph in view of their results and refer to their paper. In l. 514, you start the sentence on the Eastern Alps with "in contrast", which probably means "in contrast to the Western and Central Alps". There should not be any difference between the Western, Central and Eastern Alps for the influence of seismic anisotropy on the results of isotropic travel-time tomography because the SKS results show similar (mostly strong) delay times from west to east.

In this revised version, you added both an appendix (Fig. A1 and comment) and a supplementary information file, while you refer to appendix A1 as being part of supplementary material (l. 218), which is not the case. Please homogenize.

You refer many times to the models of Diehl et al. (2009), Zhao et al. (2016), Lippitsch et al. (2003), etc. by using shortcuts such as Diehl's, Zhao's, Lippitsch's, etc. I would suggest replacing all shortcuts by the correct references. Another way would be to introduce the shortcuts in the first citation with, for example, "Diehl et al. (2009), hereinafter referred to as Diehl".

Minor comments:
- L. 267: do you mean « coefficient value of 0.6 with respect to the beam trace » ? What is the beam trace?
- L. 296: analysis=> matrix?
- L. 359: for both coarse and fine checkerboards
- L. 365: leaking=>leakage
- L. 491: theories=>hypotheses
- L. 507: Barruol et al. (2011) was the first of a long series of papers on SKS anisotropy in the Alps; you should therefore add "e.g." before the reference.
- L. 513: signify the presence of=>be due to

- L. 585: comprises=>encompasses
- L. 588: Zhao (Zhao et al., 2016, personal communication)=> Zhao et al. (2016) (L. Zhao, personal communication)
- L. 589 : Koulakov (Koulakov, 2021)=>Koulakov et al. (2009)
- L. 592 : model=>models
- L. 595, 596, 692 : delete « already »
- L. 596 : at least
- L. 600 : (16)=>16
- L. 601 : the the=>the
- L. 603 : Alpine Tethys
- L. 614 : and (E=>and E
- L. 634-635 : the question of the role played by crustal corrections
- L. 643-644: it is favourable if the teleseismic inversion is allowed to change the structure also in the crust=>it is beneficial that teleseismic inversion is allowed to change the structure of the crust as well
- L. 646-647: delete "especially" and "therefore"
- L. 656: delete "certain"
- Caption Fig. 10: including a priori=>including the a priori; including slightly=>including the slightly
- Fig. 14: detachement=>detachment
- Caption Fig. 14: Profile distance=>Distance
- Caption Fig. 15: on longitude=>at longitude
- Caption Fig. A1: meaning of "we set the a priori variance highest at 15% of the model reference velocity"? Rephrase.

Regards
A. Paul

---

## Author Response (AR2)

Dear authors,

The revised version of your manuscript takes into account the reviewers' comments as well as mine. I therefore consider that it does not need further review by the reviewers.
I have read the revised version carefully myself, and have noted a series of corrections that you should be applied before the manuscript can be accepted for publication. A large majority of these comments deal with typos, incorrect English writing, or shortcuts in references to publications. They are listed below. Please note that line numbers refer to the authors's tracked change version, and not to the revised version.

Dear Anne Paul,

Thank you very much for reviewing our updated manuscript. Please find below our detailed comments in blue. We also made some smaller adjustments due to the input of our co-authors on the most recent version that you can find in the authors' tracked changes file. We also updated Fig. 5 which now shows azimuthal distribution of picks on a logarithmic axis instead of events, as we found this information to be much more instructive.

Kind Regards,

Marcel Paffrath

A more important comment deals with your discussion of the potential effect of seismic anisotropy on the results of your (isotropic) teleseismic travel-time tomography (lines 506-513). You write that "steeply incident teleseismic P-waves passing through these mantle regions tend to be slower than the isotropic average because they propagate in a plane perpendicular to the fast axis". This is wrong in most cases because, in first approximation, teleseismic P waves propagate in the vertical plane defined by the source-receiver back-azimuth which is generally not perpendicular to the fast-velocity direction of the SKS anisotropy. However, it is right that anisotropy related to subduction induces artificial low-velocity anomalies around slabs in seismic tomographies of subduction zones. This effect has been studied by Bezada et al. (2016, doi: 10.1002/2016GC006507). You should correct this paragraph in view of their results and refer to their paper. In l. 514, you start the sentence on the Eastern Alps with "in contrast", which probably means "in contrast to the Western and Central Alps". There should not be any difference between the Western, Central and Eastern Alps for the influence of seismic anisotropy on the results of isotropic travel-time tomography because the SKS results show similar (mostly strong) delay times from west to east.

We added the citation and changed the paragraph slightly. We now write that the teleseismic P-waves propagate nearly perpendicularly to the fast axis. We have added that anisotropy may also contribute to the negative anomalies east of 10°E, but that new results by F. Link indicate a weakening of anisotropy east of the Tauern window and we prefer an interpretation of the low velocity anomalies due to asthenospheric upwelling.

In this revised version, you added both an appendix (Fig. A1 and comment) and a supplementary information file, while you refer to appendix A1 as being part of supplementary material (l. 218), which is not the case. Please homogenize.

Okay, done

You refer many times to the models of Diehl et al. (2009), Zhao et al. (2016), Lippitsch et al. (2003), etc. by using shortcuts such as Diehl's, Zhao's, Lippitsch's, etc. I would suggest replacing all shortcuts by the correct references. Another way would be to introduce the shortcuts in the first citation with, for example, "Diehl et al. (2009), hereinafter referred to as Diehl".

Okay, done

Minor comments:
- L. 267: do you mean « coefficient value of 0.6 with respect to the beam trace » ? What is the beam trace?
  Yes we mean with respect to the beam trace. It is the stacked trace of each event.
- L. 296: analysis=> matrix?
  Okay
- L. 359: for both coarse and fine checkerboards
  Okay
- L. 365: leaking=>leakage
  Okay
- L. 491: theories=>hypotheses
  Okay
- L. 507: Barruol et al. (2011) was the first of a long series of papers on SKS anisotropy in the Alps; you should therefore add "e.g." before the reference.
  Okay
- L. 513: signify the presence of=>be due to
  Okay
- L. 585: comprises=>encompasses
  Okay
- L. 588: Zhao (Zhao et al., 2016, personal communication)=> Zhao et al. (2016) (L. Zhao, personal communication)
  Okay
- L. 589 : Koulakov (Koulakov, 2021)=>Koulakov et al. (2009)
  Okay
- L. 592 : model=>models
  Okay
- L. 595, 596, 692 : delete « already »
  Okay
- L. 596 : at least
  Okay
- L. 600 : (16)=>16
  Okay
- L. 601 : the the=>the
  Okay
- L. 603 : Alpine Tethys
  Okay
- L. 614 : and (E=>and E
  Changed to (E) (Fig. 16…)
- L. 634-635 : the question of the role played by crustal corrections
  Okay
- L. 643-644: it is favourable if the teleseismic inversion is allowed to change the structure also in the crust=>it is beneficial that teleseismic inversion is allowed to change the structure of the crust as well

Okay
- L. 646-647: delete "especially" and "therefore"
  Okay
- L. 656: delete "certain"
  Okay
- Caption Fig. 10: including a priori=>including the a priori; including slightly=>including the slightly
  Okay
- Fig. 14: detachement=>detachment
  Okay
- Caption Fig. 14: Profile distance=>Distance
  Okay
- Caption Fig. 15: on longitude=>at longitude
  Okay
- Caption Fig. A1: meaning of "we set the a priori variance highest at 15% of the model reference velocity"? Rephrase.
  Okay

Regards
A. Paul

---

## Author Response (AR3)

Dear authors,

Your paper is almost ready for publication provided that you apply the few minor corrections listed below. Once again, the line numbers refer to the authors's tracked change manuscript, and not to the last version of the manuscript.

Dear Anne Paul,

Again, thank you very much for reviewing our manuscript. Please find our detailed replies in blue below.

- L. 18 (abstract): "a short western Alpine slab whose surface trace ends about 100 km from the Penninic front"? According to the corresponding part of the discussion l. 521), you rather mean "a short western Alpine slab whose easternmost end is located at about 100 km depth beneath the Penninic front". Please correct.

Yes thank you

- L. 47-55: you cite a number of papers on seismic refraction and reflection surveys, local earthquake tomography studies, etc. but none of these lists is complete. You should then either cite all relevant papers, which are numerous, or as a stop back, start all lists with "e.g.".

Okay, done

- L. 50: you cite Fry et al. (2010) and Molinari et al. (2015), and not the first paper published on ambient-noise tomography of the Alps, which is Stehly et al. (2009, doi: 10.1111/j.1365-246X.2009.04132.x). You should at least add this one.

Okay, done

- L. 242: you probably mean "data base" rather than "data basis"

No, we mean data basis here

- Caption of Fig. 5: "will get visible WHEN (?) analysing resolution capabilities"

Yes, thanks

- L. 454, 456: rather than "Ivrea zone" (which is a surface outcrop of low-velocity rocks), you probably mean "Ivrea body" (which is the high-velocity, high-density anomalous body at 10 km depth beneath the Ivrea gravity high of the western Alps).

Yes that is right thank you

- L. 506-508: I still disagree with your argument on the influence of seismic anisotropy measured from SKS splitting analyses on the low-velocity anomalies. The SKS wave is split in a fast and a slow quasi-S waves when propagating through a transversely isotropic medium with a horizontal symmetry axis. As a consequence, both the fast-velocity and the slow-velocity directions measured from SKS splitting are horizontal. Therefore, the steeply incident teleseismic P-waves may be slower if their propagation plane is perpendicular to the fast directions, but they can be faster if it is parallel to the fast direction. As the orientation of the propagation plane depends on the source-receiver backazimuth and is highly variable, there is no single effect of reduction or increase in P-wave velocity. As analyzed by Bezada et al. (2016), the time delays of teleseismic P-waves are affected by

anisotropy with dipping symmetry axes in subduction regions. You should read this paper and be more precise on its outcomes on teleseismic tomography around subductions, and the possible implications on your results.

From seismic wave theory we know that for an anisotropic medium the velocity of the quasi P-wave depends on propagation direction, i.e. incidence angle and azimuth, while the S-wave velocity depends on polarisation and propagation direction. For a transversely isotropic medium with fast symmetry axis the P-wave velocity is highest for a propagation direction parallel to this axis and hence slower for any propagation direction oblique and in particular perpendicular to this axis. For the case of a horizontal fast symmetry axis, subvertically propagating P-waves are slower than in the isotropic case regardless of the propagation plane which is determined by the azimuth only.

Bezada et al. (2016) analyse in their paper the effect of anisotropy on isotropic teleseismic P-wave tomography. They consider three different azimuthal distributions of seismic events and find for a subhorizontal fast symmetry axis that (Sect. 4 of their paper):

"All three cases present similar artifacts indicating that, regarding anisotropy, azimuthal coverage is of secondary importance when dealing exclusively with teleseismic arrivals. The primary factor is the subvertical orientation of the teleseismic ray-paths and their consequential sampling of the slower directions of regions with subhorizontal fast axes."

This exactly confirms our view that for subhorizontal mantle flow behind the subducting Alpine slabs the subvertically propagating P-waves are slower than the isotropic average and thus could produce the low velocity regions behind the slabs in our tomographic model. We added another paragraph to the manuscript explaining the above results of Bezada et al. (2016) and their relevance for our tomography.

Kind regards,

Wolfgang Friederich and Marcel Paffrath